# Visual Prompting with Iterative Refinement for Design Critique Generation

## Abstract

Feedback is crucial for every design process, such as user interface (UI) design, and automating design critiques can significantly improve the efficiency of the design workflow. Although existing multimodal large language models (LLMs) excel in many tasks, they often struggle with generating high-quality design critiques—a complex task that requires producing detailed design comments that are visually grounded in a given design's image. Building on recent advancements in iterative refinement of text output and visual prompting methods, we propose an iterative visual prompting approach for UI critique that takes an input UI screenshot and design guidelines and generates a list of design comments, along with corresponding bounding boxes that map each comment to a specific region in the screenshot. The entire process is driven completely by LLMs, which iteratively refine both the text output and bounding boxes using few-shot samples tailored for each step. We evaluated our approach using Gemini-1.5-pro and GPT-4o, and found that human experts generally preferred the design critiques generated by our pipeline over those by the baseline, with the pipeline reducing the gap from human performance by 50% for one rating metric. To assess the generalizability of our approach to other multimodal tasks, we applied our pipeline to open-vocabulary object and attribute detection, and experiments showed that our method also outperformed the baseline.

## 1 Introduction

Critiques are essential for design, providing feedback to help designers improve their work (Duan et al., 2024a; Wang et al., 2021; Xu et al., 2014). However, obtaining design critiques is often costly and time-consuming, hindering the design process. Hence, automating design critiques has become an important goal in many design fields. In this paper, we focus on automating critiques for user interface (UI) design—a prevalent task in industry that directly impacts the user experience (Stone et al., 2005). Obtaining UI design feedback typically requires expert reviews or user testing with target end users, which may be expensive and not always readily available. This makes automated critique extremely valuable, as it can provide instant feedback for designers to quickly iterate on (Duan et al., 2024a). Furthermore, automated design feedback can serve as a reward function for automated UI generation, which has started to gain traction (Gajos et al., 2010; Gajjar et al., 2021).

UI design critique is often complex and open-ended, involving feedback that covers multiple dimensions of the design (e.g., aesthetics and usability) (Nielsen & Molich, 1990; Hartmann et al., 2008) and addresses both the overall design and specific problematic regions of the UI, based on design principles or guidelines. This makes automated UI critique a very challenging task. Given a UI screen and a set of design guidelines, the model needs to understand the screen, reason with UI design principles to detect violations in the UI design (both semantically and spatially), and then explain and contextualize the feedback in the way that human designers can understand and act upon (Duan et al., 2024b) (Figure 1). Essentially, automated UI design critique is a challenging task that presents an opportunity to develop various multimodal capabilities in models.

Multimodal LLMs have made tremendous progress in a variety of multimodal tasks, such as visual question answering (VQA) and visual understanding, due to their extensive knowledge and generalization capabilities. Although multimodal LLMs appear to be readily usable for design critique, a multimodal task, there remains a significant gap in quality between the feedback generated by

these LLMs compared to that of human design experts (Duan et al., 2024b). In addition, multimodal LLMs often struggle with achieving accurate visual grounding (Duan et al., 2024b; Dorkenwald et al., 2024), making it difficult for them to mark relevant regions in the UI screenshots, which is crucial for contextualizing feedback for designers (Duan et al., 2024b).

Recent advances in prompting techniques have improved both visual grounding and text generation performance. For example, Fang et al. (2024) introduced a visual prompting technique that adds visual markers to an image, which helps multimodal LLMs better ground objects. Madaan et al. (2023); Xu et al. (2024a) proposed a method called *iterative refinement*, where an LLM's output is repeatedly refined by itself or another model until the output is deemed correct. *iterative refinement* has been shown to improve the LLM's performance for text-only tasks like code optimization and machine translation. Building on these prompting methods, we develop a novel technique for UI design critique generation (Figure 1). Our approach iteratively refines both design comment text and their corresponding bounding boxes, utilizing visual prompting to assist in bounding box generation and refinement. For iterative refinement of bounding boxes, we introduce a novel technique that displays a zoomed-in patch of the bounding box candidate to help the refinement process. Our approach is implemented through an architecture that coordinates multiple multimodal LLMs (Figure 2).

We evaluated our pipeline for UI critique using UICrit, a public dataset (Duan et al., 2024b), with two state-of-the-art multimodal LLMs: Gemini-1.5-pro (Team et al., 2024) and GPT-4o (OpenAI et al., 2024). Our experiments demonstrated that the pipeline consistently improved the design feedback output across both models, on both automatic metrics and human expert evaluation. To assess the broader applicability of our method to other multimodal tasks, we tested it on open-vocabulary object and attribute detection, where it consistently increased the mAP by up to 9.1. These experiments demonstrate the potential of our method to be a useful technique in the broader scope of tasks, beyond design critique generation, pushing the boundary of what prompting can achieve for complex multimodal tasks. Our paper makes the following contributions:

- A modular multimodal prompting framework that orchestrates six LLMs (Figure 2) for generation, refinement, and validation of design critiques, which takes in an image and a task prompt, and generates a list of text items visually grounded in the image.

- A set of LLM prompting techniques for iterative refinement of both text and bounding boxes that ground the text within the image. We introduce a technique for visual grounding refinement, where we include a zoomed-in patch around the bounding box candidate (Figure 3) in the prompt to assist in fine-grained visual grounding.

- Extensive experiments with the proposed prompting framework on UI design critique, a challenging multimodal task, and a study of its performance on open vocabulary object and attribute detection. These experiments showed that our pipeline consistently outperformed the baseline methods across two distinct multimodal tasks and domains.

## 2 RELATED WORK

### 2.1 AUTOMATED UI DESIGN CRITIQUE WITH LLMS

Prior work have studied the capabilities of using LLMs for UI design critique. Duan et al. (2024a) explored the performance of zero-shot (text-only) GPT-4 in critiquing UI mockups, using a JSON representation of the UI. They identified gaps between the feedback capabilities of general-purpose LLMs and human experts. To address this, Duan et al. (2024b) collected a dataset (UICrit) consisting of human-annotated UI design critiques (grounded within UI screenshots via bounding boxes) for UI screens that could be applied to train general purpose LLMs. Their UI design critique model takes in a UI screenshot and outputs critiques grounded in screenshot regions. Their method showed a significant improvement in LLM-generated feedback with just few-shot sampling from UICrit, although the feedback quality still falls short of human experts. Similarly, Wu et al. (2024) generated a synthetic dataset of UI design comments and trained a CLIP model (Radford et al., 2021) to assess UI designs. We apply our approach to the design critique task, which augments the method of Duan et al. (2024b) by incorporating iterative refinement of both the design comments and their corresponding bounding box positions on the UI screen.

## 2.2 PROMPTING LLMs WITH ITERATIVE REFINEMENT

Iterative refinement on LLM output has been explored in prior studies, as a means to improve LLM performance. Madaan et al. (2023) developed an approach called "SELF-REFINE", where a single LLM generates an initial output and then iteratively provides feedback on its own output and revises the the output based on the feedback. They applied this technique across a diverse set of tasks, such as math reasoning and dialogue response, and found that SELF-REFINE resulted in an 20% average performance gain. Similarly, Zhou et al. (2023) utilized this iterative self-refinement technique on long-horizon sequential task planning in robotics, leading to higher success rates. However, Xu et al. (2024b) found that LLMs often exhibit *self-bias* (i.e. a tendency to favor its own generated output) during self-refinement across a variety of tasks and languages. To account for this, Xu et al. (2024a) developed "LLMRefine", a method for text generation that uses a separate model to provide detailed feedback, along with a simulated annealing method to iteratively refine the LLM's output. We also utilize iterative refinement in our pipeline, and we extend this method to multimodal tasks by refining both text and bounding boxes that associate the text with relevant regions in the image. Following the method in LLMRefine, we use separate LLMs for generation and refinement to prevent self-bias.

## 2.3 MULTIMODAL TASKS

Previous work has investigated a variety of grounded multimodal tasks using LLMs, where an LLM takes in a visual input (such as an image) and generates outputs that are connected to specific objects, regions, or attributes within the visual input. Liu et al. (2023) introduced Grounding DINO, a transformer-based model that supports open-vocabulary object detection and can identify arbitrary objects within an image. However, it struggles with complex queries involving multiple objects and intricate spatial relationships. To address this limitation, Zhao et al. (2024) developed LLM-Optic, which uses an LLM to break down complex queries into specific objects, employs Grounding DINO to detect candidate objects, and finally uses a multimodal LLM to select the most suitable objects for the query. Beyond object detection, Bravo et al. (2023) introduced open vocabulary object and attribute detection, which identifies and grounds both objects and their corresponding attributes in an image in a open vocabulary setting. In robotics, multimodal LLMs were used to help systems understand the physical world. Fang et al. (2024) introduced MOKA, which utilizes multimodal LLMs to solve complex robotic manipulation tasks by breaking them into multiple steps. Their approach incorporates visual prompting, where visual markers are added to the image, to aid in object grounding as part of the robot's step-by-step instructions. Chen et al. (2024) examined the capabilities of multimodal LLMs for evaluation across three tasks: pair comparison, scoring, and ranking. They found that while LLMs performed well on pair comparison, they struggled with the other tasks, suggesting that further improvements are needed before LLMs can be reliable validators. Visual grounding is a vital component of our method, and we utilize visual prompting to enhance bounding box generation and refinement. Although multimodal validation has limitations, our ablation studies indicate that incorporating it to validate the generated text and bounding boxes generally improved performance.

## 3 TASK

*UI design critique generation* was first proposed as a grounded multimodal task by Duan et al. (2024b). The model takes in a UI screenshot and a set of design guidelines as input and outputs a list of design critiques. Each design critique comprises two components: a text comment that identifies a specific issue in the UI and a bounding box that highlights the relevant region of the screenshot (see Figure 1). For example the text comment might state "*The expected standard is to use clear contrast for readability. In the current design, the label 'Best' is difficult to see on the image due to its high transparency. To fix this, reduce the transparency of the box and apply a solid color so that the text 'Best' is readable.*") and the bounding box will enclose the orange 'Best' tag in the UI screenshot in 1.

As discussed earlier, the UI design critique task is particularly challenging because the model must understand and apply UI design principles to identify design issues in the screenshot. Furthermore, determining the exact region of the screen (i.e., the bounding box) for a comment is not always straightforward. For example, a comment might note that the text in the UI has poor contrast with the background, but not specify which text element is problematic, requiring the model to identify

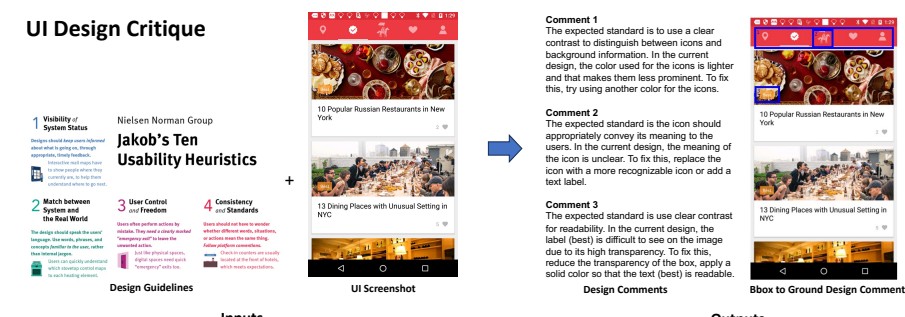

Figure 1: Illustration of the UI Design Critique Task, which takes in a UI screenshot and a set of design guidelines and outputs a list of design comments with corresponding bounding boxes (Bbox).

the problematic elements and also determine their bounding box. While our focus in this paper is on UI design critique, our task is representative of many multimodal tasks that require visually grounded text generation.

## 4 METHOD

We developed a prompting pipeline that uses multiple LLMs to generate UI design critiques. It consists of six distinct LLMs that are organized into three modules: *Text Generation & Refinement*, *Validation*, and *Bounding Box Generation & Refinement*. These modules communicate with each other to complete the task. Figure 2 illustrates the workflow of the pipeline, showing the main inputs and outputs of each LLM, which are numbered by the order of execution. We break down the entire task into separate generation and refinement steps for both text and bounding boxes, as decomposing complex tasks has been shown to improve performance (Khot et al., 2023).

As shown in the figure, the LLM output of each step is conditioned on that of the previous step. Since Bounding Box Generation & Refinement is conditioned on the text predictions, and text refinement, in turn, is conditioned on the bounding box predictions, we introduce the Validation module between the Text and Bounding Box modules to ensure that each refinement step is based on more accurate inputs. Additionally, each LLM is provided with targeted few-shot examples to improve its accuracy, as well as a text prompt containing specific instructions for that step, which is derived from the input task prompt. To provide as much guidance as possible, we included the UI design guidelines in the input task prompt, which are also included in the instructions prompts for relevant steps. The specific inputs, outputs, and few-shot examples for each LLM are detailed in the following sections, and the instructions prompt for each step can be found in Appendix A.3.

**Text Generation LLM (TextGen)** The pipeline begins with the TextGen LLM that takes an image and its instructions prompt (derived from the task prompt) as input, and generates a list of un-grounded text items (design comments) for the image. We decided to start with text generation and condition the bounding box generation on the generated text, instead of the other way around. This decision is based on our observation that for design critique, LLMs tend to perform poorly on visual grounding from scratch (i.e., without guidance from text), which makes the subsequent refinements much more error-prone.

**Text Filtering LLM (TextFilter)** To reduce the chance of bounding box generation being conditioned on incorrect text items (i.e., incorrect design comments), we add an additional filtering step to remove invalid or irrelevant text items. The TextFilter LLM takes as input a list of generated text items from TextGen, along with the image, and outputs a filtered list of valid text items. While previous studies (Chen et al., 2024; Shankar et al., 2024) have shown that LLMs may not always be reliable evaluators, Liu et al. (2024) demonstrated that few-shot examples can improve performance. We designed few-shot examples for TextFilter by injecting invalid items into a correct list of text items, using this augmented list as input and the original correct list as the expected output. This illustrates how to filter out invalid items.

**Bounding Box Generation LLM (BoxGen)** The BoxGen LLM generates bounding boxes based on the filtered text items from TextFilter. The LLM takes in one text item at a time, as well as

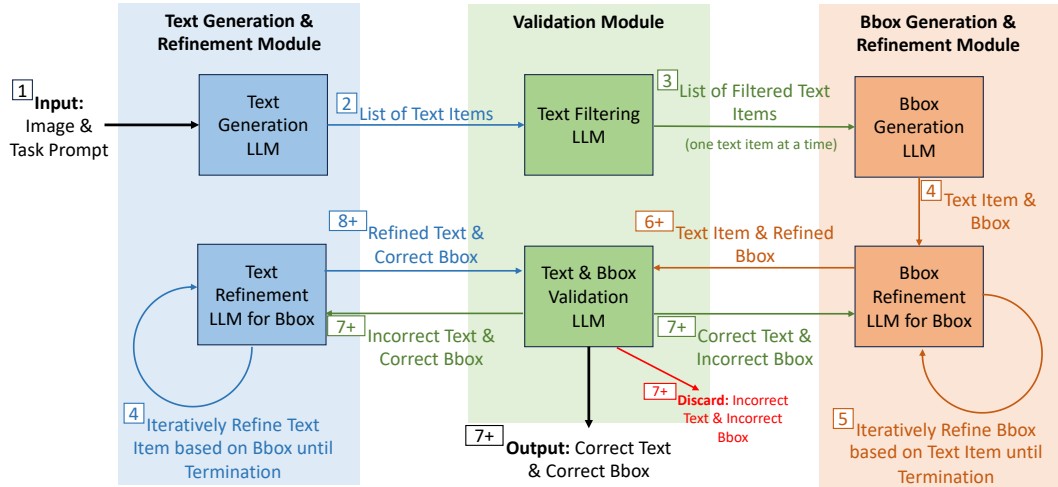

Figure 2: The figure illustrates our prompting pipeline, which takes an image and a task prompt as input and outputs text items with their corresponding bounding boxes on the image. The pipeline consists of six distinct LLMs, organized into three modules: Text Generation and Refinement, Validation, and Bounding Box (Bbox) Generation and Refinement. Targeted few-shot examples are provided for each LLM. The main inputs and outputs for each LLM are shown, and Section 4 details all the inputs, outputs, and few-shot examples for each LLM. Each input/output is numbered with their order of generation, and numbers with a '+' indicate multiple iterations of input/output.

the image, and predicts a relevant region on the image via bounding box coordinates. Following the visual prompting technique from Duan et al. (2024b), we augment the screenshot by adding coordinate markers along its edges (Figure 3) to help the LLM associate coordinates with specific regions in the screen.

**Bounding Box Refinement LLM (BoxRefine)** To avoid self-bias during iterative refinement (Xu et al., 2024b), we use a separate LLM to iteratively refine the generated bounding box from the previous step. The BoxRefine LLM takes in several inputs, as shown in Figure 3. Similar to Box-Gen, BoxRefine takes in the coordinate-marker enhanced screenshot image and a filtered text item. Additionally, BoxRefine takes in the bounding box coordinates that was predicted by BoxGen, and a close-up view of the image region specified by the predicted bounding box coordinates. In this zoomed-in image patch, the bounding box is displayed as a blue box, with some surrounding region of the box included for additional context. The zoomed-in image patch also has coordinate markers along the edges to help the LLM refine the bounding box coordinates based on this close-up view.

The LLM assesses the quality of the current bounding box based on all these inputs. If the bounding box is deemed accurate by the BoxRefine LLM, the iterative refinement process terminates. Otherwise, the LLM returns the refined coordinates, which are then re-evaluated by the LLM. This process is repeated until the LLM either confirms the bounding box as correct or the maximum number of iterations is reached. Previous work (Madaan et al., 2023) has shown that the history of refinements provides helpful information. Thus, we include the history of the LLM's refinements for the input bounding box as an input at each iteration, which enables the model to learn from past adjustments. Few-shot examples are generated by creating a synthetic refinement sequence with gradually reduced noise in the perturbation of a sampled bounding box's coordinates. Algorithm 1 in the Appendix details our methods for bounding box perturbation and the generation of few-shot examples for bounding box refinement.

**Text & Bounding Box Validation LLM (Validation)** After determining the bounding box for the text item, the Validation LLM determines if the bounding box and text are correct and can be used in the final output, or if they require further refinement. The Validation LLM takes as input the entire image, a zoomed-in image patch for the proposed region specified by the bounding box, and the text item, and assesses the accuracy of critique generation as one of the following:

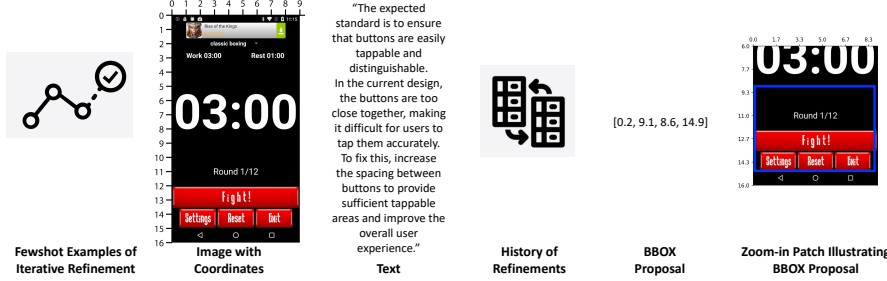

Figure 3: An example of the inputs to the Bounding Box Refinement LLM.

1. *Both Text & Box Correct:* Both the bounding box and the text item are accurate, and the pair is returned. The pipeline moves onto the next text item in the filtered list.
2. *Incorrect Text:* The bounding box correctly identifies a region in the UI screenshot with design issues, but the text item is incorrect (e.g., does not adequately describe the design issues in the region). The pair is sent to the TextRefine LLM for text refinement.
3. *Incorrect Bounding Box:* The text item is correct (e.g., describes a valid design issue in the UI screenshot), but the bounding box is incorrect (e.g., does not accurately enclose the region described in the critique). The bounding box and text item are sent back to the BoxRefine LLM for further refinement of the bounding box.
4. *Both Incorrect:* Both the text and the bounding box are incorrect. The pair is discarded and the pipeline moves onto the next text item in the filtered list.

Few-shot examples are generated differently for each case; the bounding box is perturbed for the Incorrect Bounding Box case (Algorithm 1 in the Appendix), the text item is perturbed for the Incorrect Text case, and both the bounding box and text are perturbed for the Both Incorrect case. In addition, text and bounding box pairs that are sent for further refinement are sent back to this LLM for validation, after they have been refined.

**Text Refinement LLM (TextRefine)** The TextRefine LLM is used to refine incorrect text items conditioned on bounding boxes that correctly identify relevant regions in the image, as determined by the Validation LLM. This iterative refinement process mirrors the bounding box refinement procedure. The LLM takes as input the entire image, a zoomed-in image patch focused on the bounding box, and the text item, and refines the text iteratively until it determines that the text is accurate for the region shown in the bounding box. Few-shot examples are generated either by perturbing the text (if possible) or by selecting irrelevant text items from the few-shot dataset and then ranking them by increasing semantic similarity to simulate the refinement process. The refined text item and bounding box are then returned to the Validation LLM.

## 5 EXPERIMENTS

### 5.1 DATASET

We used the UICrit dataset[1], collected by Duan et al. (2024b), to evaluate our pipeline for the design critique task. Each UI screenshot in this dataset was annotated by three experienced human designers, providing feedback that includes a list of text-based design critiques with their corresponding bounding boxes, numerical ratings for usability, aesthetics, and overall design quality, as well as a description of what the screen is designed for. The dataset contains a total of 11,344 design critiques for 1,000 screenshots. For evaluation, we used the UI screenshots from UICrit as input images, included the three sets of design guidelines used by Duan et al. (2024b) in the task prompt, and evaluated the model's output against the comments and bounding boxes of the screen from the dataset (depending on the experiment). For few-shot examples, we sampled from a split of UICrit that is separate from the examples used for the evaluation. The few-shot sampling methods used at each step is detailed in Appendix A.2.1.

---

[1]https://github.com/google-research-datasets/uicrit

Table 1: IoU values from the Ablation study on the different components of bounding box generation. IR stands for Iterative Refinement, and VP stands for Visual Prompting.

| Methods | UI Critique IoU $\uparrow$ | |
| --- | --- | --- |
| | Gemini$_{1.5tn}$ | GPT$_{4tn}$ |
| Zero-shot | 0.120 | 0.233 |
| Zero-shot, VP | 0.180 | 0.249 |
| Few-shot, VP | 0.267 | 0.319 |
| Few-shot, VP, Zero-shot IR | 0.279 | 0.319 |
| **Few-shot, VP, Few-shot IR** | **0.357** | **0.345** |

## 5.2 BASELINE

We used the few-shot pipeline developed by Duan et al. (2024b) for their UI critique task as the baseline. Their pipeline consists of the Text Generation LLM (Figure 2) with few-shot sampling, followed by an LLM for bounding box generation that uses visual prompting (i.e., coordinates marked on the screenshot edges) without few-shot examples.

## 5.3 IMPACT OF VISUAL PROMPTING & ITERATIVE REFINEMENT ON VISUAL GROUNDING

Table 1 presents an ablation study on the different components of the Bounding Box Generation and Refinement module (Figure 2), which illustrates the impact of visual prompting and iterative refinement on the visual grounding accuracy of two state-of-the-art multimodal LLMs: Gemini-1.5-pro and GPT-4o. For this evaluation, the module is given a UI screenshot and one of its comments from UICrit. Its output bounding box is evaluated against the ground-truth bounding box of that comment in UICrit by computing their IoU. The module consists of two LLMs (BoxGen and BoxRefine), and the BoxRefine LLM was only used for the conditions with iterative refinement (i.e., the last two rows of the table).

For Gemini-1.5-pro, each enhancement led to an improvement in the average IoU, with the final setup (used in our pipeline) achieving an average IoU nearly three times higher than zero-shot and almost double that of zero-shot with visual prompting, which was used in the baseline (Duan et al. (2024b)). For GPT-4o, improvements were seen at each step, except for zero-shot iterative refinement; when no few-shot examples were provided in the refinement prompt, GPT-4o did not refine any of the input bounding boxes. Additionally, while GPT-4o had better zero-shot performance, its IoU for the final setup was slightly worse than that of Gemini-1.5-pro. Nevertheless, iterative visual prompting led to substantial performance gains over zero-shot prompting for both LLMs, indicating that iterative visual prompting significantly enhances bounding box estimation.

## 5.4 PIPELINE ABLATION STUDY AND QUALITATIVE ANALYSIS

Table 2 presents the results of the ablation study for UI design critique for both LLMs, as well as the results for the baseline setup and multimodal Llama-3.2 11b (Dubey et al., 2024), which has been finetuned on the training split of UICrit for three epochs. Since UI design critique is open-ended, UICrit does not contain all the ground-truth design comments for each UI screenshot. Hence, we evaluated comment generation by computing the cosine similarity of sentenceBERT embeddings with each comment in the dataset for the UI screenshot and selecting the highest one ("Comment Similarity" in Table 2). The IoU was estimated by comparing the predicted bounding box with that of the most semantically similar comment ("Estimated IoU" in Table 2). The estimated IoU values are lower than those in Table 1, where the IoU was calculated directly from the input comments' corresponding bounding boxes in UICrit. The estimated IoU is lower because it uses the bounding box of the most semantically similar comment in the dataset instead, which may not precisely match the comment for which the bounding box was generated.

Each step of the pipeline incrementally improved the comment similarity and estimated IoU for both LLMs. While GPT-4o and Gemini-1.5-pro showed similar values in terms of comment similarity, GPT-4o achieved a higher estimated average IoU. GPT-4o's advantage could be due to its significantly larger size—nearly three times as many parameters as Gemini-1.5-pro. The complete pipeline also outperforms the baseline in both comment similarity and estimated IoU. Note that the comment

Table 2: Ablation study of the different steps of our pipeline on UI design critique. IR stands for Iterative Refinement. Note that we combine the results of the Validation step with results from the additional iterative refinement steps for bounding box and text. This is because these additional refinements are applied to a much smaller subset; specifically, only the pairs identified as having incorrect text or incorrect bounding boxes during the Validation step. We also include results from the baseline setup and finetuned Llama-3.2 11b.

| Pipeline Steps | Comment Similarity ↑ | | Estimated IoU* ↑ | |
|---|---|---|---|---|
| | $Gemini_{1.5tn}$ | $GPT_{4tn}$ | $Gemini_{1.5tn}$ | $GPT_{4tn}$ |
| Text Generation | 0.651 | 0.680 | N/A | N/A |
| + Text Filtering | 0.694 | 0.692 | N/A | N/A |
| + Bbox Generation | 0.694 | 0.692 | 0.153 | 0.244 |
| + IR of Bbox | 0.694 | 0.692 | 0.173 | 0.259 |
| + Validation, IR of Text & Bbox | 0.702 | 0.701 | 0.199 | **0.275** |
| Baseline (Duan et al., 2024b) | 0.651 | 0.680 | 0.176 | 0.257 |
| Finetuned Llama-3.2 11b | **0.842** | | 0.230 | |

similarity for the baseline is identical to that of the 'Text Generation' row. This is because both the pipeline and baseline start with TextGen, so we used the same initial comments from TextGen for both conditions for easier comparison. Fine-tuned Llama-3.2 achieves higher comment similarity than the pipeline, but its estimated IoU falls between those of Gemini-1.5-pro and GPT-4o for the complete pipeline.

We conducted a qualitative analysis of the outputs from the pipeline, baseline, and finetuned Llama-3.2, finding the pipeline outputs helpful comments with reasonable bounding boxes (more often than not) and generally outperforms the baseline and finetuned Llama. Compared to baseline, it reduces the generation of invalid and generic comments, while producing bounding boxes that are tighter, more specific, and closer to the target region. However, the pipeline sometimes eliminates valid comments. Also, we found that the baseline often generates very large bounding boxes that cover the majority of the screen. This would decrease the chance of the IoU being zero, which may have inflated its estimated IoU. We found that finetuned Llama only generated a very limited set of critiques, while our pipeline generates a considerably more diverse set of comments. Although finetuned Llama generally had better visual grounding, the bounding boxes tend to be larger and less specific. Section A.4.1 (Appendix) provides detailed results and example outputs. Section A.4.2 presents qualitative results and outputs for out-of-domain UIs (e.g. websites), demonstrating that our pipeline can still generate helpful feedback. Finally, Section A.5 includes a cost analysis of our pipeline and also contains example visualizations of bounding box and comment refinements.

## 5.5 HUMAN EVALUATION

Due to the open-ended nature of UI design critique, UICrit does not have the complete set of ground-truth design comments for each UI screen. Hence, we recruited human design experts to assess the validity of the feedback generated by our pipeline. For comparison, the experts also rated the comments generated by the *baseline* setup and human annotated comments from UICrit. We used the same procedure devised by Duan et al. (2024b), where each design comment was rated as invalid, partially valid and valid, and the set of design comments from each condition was ranked as a whole, based on overall quality and comprehensiveness. Unlike the method used by Duan et al. (2024b), where participants rated both comment quality and bounding box accuracy together, our evaluation presented participants with a screenshot marked with a ground-truth bounding box (determined and agreed upon by the authors) and asked them to rate the validity of the comment only for that region. This is to ensure a more rigorous and standardized approach to evaluate bounding box accuracy and a more focused evaluation on comment quality. See Section 5.5 (Appendix) for more details on the study method. Table 3 shows the average comment rating, the average comment set rank, and the average IoU for each of the three conditions for Gemini-1.5-pro's output. We used the established ground-truth bounding boxes from comments rated as valid or partially valid to compute the IoU with predicted bounding boxes. For the "human" condition, the IoU was not computed as we displayed the bounding boxes from UICrit. The average Fleiss Kappa inter-rater reliability score (Fleiss et al., 1971) amongst the participants was 0.22 for comment quality and 0.29 for comment set ranking, indicating fair agreement.

Table 3: Human expert ratings on UI design comments generated by Gemini-1.5-pro, and IoU of the generated bounding boxes for human validated comments.

| Methods | Comment Quality ↑ | Comment Set Rank ↓ | BBox IoU ↑ |
|---|---|---|---|
| Baseline (Duan et al., 2024b) | 0.45 | 2.3 | 0.423 |
| Our Pipeline | 0.47 | 2.0 | **0.451** |
| **Human** | **0.56** | **1.7** | N/A |

Across all the metrics, the pipeline outperformed the baseline, while human annotations remain the best. Interestingly, the average comment quality rating for human feedback was lower than expected, which may be attributed to the subjective nature of design critique (Nielsen & Molich, 1990) and the variability in dataset quality, potentially due to UICrit's annotators' limited design experience (Duan et al., 2024b). While the gap between our pipeline and the baseline is modest, it still closes 22% of the gap between the baseline and human condition. Notably, the average comment set rank of our pipeline is positioned midway between the human and baseline setups. The comment set from our pipeline was preferred over the baseline's 58% of the time and was even favored over the human condition 38% of the time.

# 6 GENERALIZATION TO OTHER TASKS

Our pipeline can be applied to other multimodal LLM tasks that involve generating visually grounded text. To assess if its performance enhancement generalizes to other tasks, we evaluate our pipeline on an existing vision-language modeling benchmark: Open Vocabulary Object and Attribute Detection (Bravo et al., 2023).

## 6.1 OPEN VOCABULARY OBJECT AND ATTRIBUTE DETECTION

Open vocabulary object and attribute detection, developed by Bravo et al. (2023), involves detecting objects and their associated attributes, along with bounding boxes marking their locations in the image (see Appendix A.1). During inference, the model is given a set of object classes and attributes to identify, including classes and attributes that were not seen during training, which tests the model's ability to generalize to novel object classes and attributes (i.e., "open vocabulary"). Bravo et al. (2023) evaluated both attribute detection (OVAD) and object detection (OVD) in this open vocabulary setting. They collected a dataset[2] of human annotated object classes and attributes for 2,000 images from the MS COCO dataset (Lin et al. (2014)), including 80 object classes and 117 attribute categories. The object classes are divided into base and novel categories, with only the base classes seen during training. We used this dataset to evaluate our pipeline on this task. The task involves taking an image as input, along with a task prompt specifying the object and attribute classes. The output is evaluated against the ground truth object and attribute annotations. To meet the open-vocabulary criterion of this task, we sampled few-shot examples from the base classes only, from a split of their dataset, but used all the classes for evaluation. Appendix A.2.2 describes the fewshot sampling strategy in more detail.

## 6.2 COMPARISON WITH BASELINE

Table 4 presents the results of the ablation study for open-vocabulary object and attribute detection, using both Gemini-1.5-pro and GPT-4o. We used the same baseline described in Section 5.2, as it can also be applied to this task. We followed the evaluation method of Bravo et al. (2023), calculating the mean average precision (mAP) across all attribute (OVAD) and object categories (OVD). The predicted text and corresponding bounding box were matched with the ground truth by selecting the bounding box with the highest IoU, with a minimum threshold of 0.5, and comparing the object categories and attribute classes.

Our approach outperformed the baseline mAP for OVAD by 2.5 and OVD by 4.6 with Gemini-1.5-pro, and by 2.2 for OVAD and 9.1 for OVD with GPT-4o. The larger performance gain for OVD may be due to the fact that it is a simpler task, with only 80 object categories compared to 117 attribute

---

[2] https://ovad-benchmark.github.io/

Table 4: Ablation study on the open vocabulary attribute detection (OVAD) and object detection (OVD) for Gemini-1.5-pro and GPT-4o. IR stands for Iterative Refinement. Note that bounding boxes are required for computing the mAP, so we combined the results for the text generation, text filtering, and bounding box generation steps. Similar to Table 2, we combined the results of the Validation step with the additional iterative refinements of the bounding box and text.

| Pipeline Steps | OVAD mAP $\uparrow$ | | OVD mAP $\uparrow$ | |
|---|---|---|---|---|
| | Gemini$_{1.5tn}$ | GPT$_{4tn}$ | Gemini$_{1.5tn}$ | GPT$_{4tn}$ |
| Text Generation + Filtering + BBox | 11.3 | 13.1 | 13.1 | 15.8 |
| + IR of BBox | 12.6 | 14.0 | 15.8 | 17.8 |
| **+ Validation, IR of Comment & BBox** | **13.6** | **15.1** | **15.8** | **20.2** |
| Baseline | 11.1 | 12.9 | 11.2 | 11.1 |

categories, and attributes are often more nuanced and harder to detect. Additionally, GPT-4o slightly outperformed Gemini-1.5-pro, likely due to its much larger size. However, our pipeline still falls short of the fine-tuned model from Bravo et al. (2023) (mAP 18.8 for OVAD and 39.3 for OVD).

## 7 DISCUSSION

Our pipeline outperforms the baseline for UI critique in both comment quality and grounding accuracy, based on automatic metrics (e.g., IoU) and human expert ratings; its feedback was also more often preferred by human experts. This implies that the design feedback generated by our pipeline is more useful for human designers. Its performance improvement also generalizes to open-vocabulary object and attribute detection, suggesting the technique could be potentially applied to enhance other grounded multimodal LLM tasks.

While our technique outperforms the baselines for open vocabulary object and attribute detection, it falls short of the fine-tuned LLMs from Bravo et al. (2023). This is expected, since our pipeline does not involve parameter-tuning, whereas their fine-tuned LLMs were trained on significantly more data than the few-shot examples provided to our model. For design critique, our pipeline generates a significantly more diverse set of critiques compared to finetuned Llama 3.2, potentially making our pipeline more useful in practice. However, our pipeline still has room for improvement when compared to human expert design feedback. Despite its performance gap with human critique (which are expensive to acquire), the generalizability of our pipeline and its consistent performance improvement over the baseline demonstrate its potential to be a versatile and resource-efficient solution for improving multimodal LLM performance across different tasks and domains.

A reason for the performance gap could be that the LLM-based validation steps are not fully accurate (Shankar et al., 2024; Chen et al., 2024), which could lead to incorrect judgement of the bounding box and/or text accuracy. Future work can improve the validation step with better prompting strategies, or look into a human-in-the-loop approach, where human experts validate or refine the text and bounding boxes. The human-in-the-loop validation could both improve the immediate quality of the output and help the system learn from human inputs over time via targeted few-shot examples. This step can be integrated into a design tool where designers validate or refine the feedback, so the model learns to provide more accurate and personalized design critiques over time.

## 8 CONCLUSION

We introduce a novel prompting pipeline that improves both the quality and visual grounding of automated UI design critique by using visual prompting and iterative refinement of both text and bounding boxes. Our approach outperformed the baseline in human evaluations, generating higher quality comments with more accurate visual grounding. Additionally, we demonstrated the generalizability of our technique through performance gains in open-vocabulary object and attribute detection, suggesting its potential to enhance other grounded multimodal tasks. While our method has limitations, it offers a versatile and resource-efficient solution for improving multimodal LLM performance across various tasks and domains.

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

## A  APPENDIX

### A.1  OPEN VOCABULARY OBJECT AND ATTRIBUTE DETECTION TASK

Open vocabulary object and attribute detection, developed by Bravo et al. (2023), is a benchmark task that involves detecting objects and their associated attributes, along with bounding boxes marking their locations in the image. Figure 4 shows an example for the Open Vocabulary Object and Attribute Detection Task. For further details about the task and the dataset, see the original paper (Bravo et al., 2023).

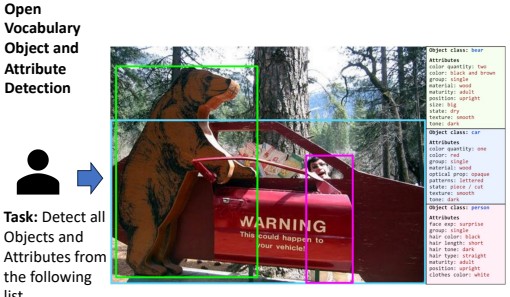

Figure 4: Illustration of the Open Vocabulary Object and Attribute Detection Task. The example output is taken from Bravo et al. (2023).

### A.2  FEW-SHOT SAMPLING METHODS FOR BOTH TASKS

#### A.2.1  UI DESIGN CRITIQUE

For both design comment generation and filtering, we sampled UI screenshots and corresponding comments based on UI task and visual similarity from a split of UICrit, following the best-performing sampling method from Duan et al. (2024b). We used CLIP (Radford et al., 2021) to

generate joint task and screenshot embeddings, and cosine similarity to determine relatedness. For filtering, we augmented the dataset's comments with LLM-generated comments deemed incorrect by annotators (Duan et al. (2024b)). For bounding box generation, refinement, and subsequent steps that operate on individual comments, we sampled few-shot examples by selecting the most semantically similar comments and their corresponding bounding boxes from a split of UICrit. We used sentenceBERT (Reimers & Gurevych, 2019) to embed the comment text for similarity ranking. For validation, few-shot examples of invalid comments were selected from incorrect comments that were marked by dataset annotators, or from irrelevant comments from other UIs. Finally, for text refinement, multiple invalid comments were selected, following the process described earlier, and then sorted by increasing cosine similarity to simulate the comment refinement process.

For bounding box refinement, we considered another technique to generate fewshot examples. This technique involves selecting the first bounding box location based on visual similarity of the region it contains in the fewshot UI to that of the region contained by the input bounding box proposal of the input screenshot. This bounding box is then gradually moved closer to the ground truth bounding box for the fewshot UI to simulate the refinement process. However, we found that the simpler approach of randomly perturbing the bounding box actually gave better results (IoU 0.357 (random perturbation, from Table 1) vs 0.333 (visual similarity match)).

### A.2.2 OPEN VOCABULARY OBJECT AND ATTRIBUTE DETECTION

For text generation (i.e., category and attributes) and filtering, we sampled images based on the semantic similarity of their CLIP embeddings. Negative text samples for the filtering step were generated by sampling irrelevant text from other images. For bounding box generation, refinement, and subsequent steps applied to individual text items, we sampled few-shot examples by selecting the most semantically similar text items and their corresponding bounding boxes from a split of their annotated dataset. We used sentenceBERT Reimers & Gurevych (2019) to embed the text items for similarity ranking. For validation, invalid text examples were perturbed by randomly swapping the category or attributes, or by deleting or adding attributes. Similarly, for text refinement, few-shot examples were generated by perturbing the text in decreasing amounts.

### A.3 INSTRUCTIONS PROMPTS FOR PIPELINE

We provide the instructions prompt for each step of the pipeline for the UI Critique Task.

```
Text Generation: For these sets of guidelines: [Guidelines]. Please find
    all the guideline violations in the UI provided. For violation found,
     please provide an explanation that includes these three things: 1.
    the expected standard (i.e. what good design should look like), 2.
    the gap between the current design and the expected standard (i.e.
    the critique for the design), and 3. how to fix the issue in the
    current design. For formatting each violation, please include these
    three things in separate sentences. For the expected standard (#1),
    start the sentence with 'The expected standard is that...'. For the
    gap (#2), start the sentence with 'In the current design, ...', and
    for how to fix the design (#3), start the sentence with 'To fix this
    ...'. Please end each violation explanation with two newline
    characters (\n\n). Please be specific in your violation explanations,
     making sure to refer to specific UI elements and groups in the UI.
    After determining all guideline violations, please also share any
    other design feedback you have for the UI and follow the same format
    of providing the expected standard, the critique for the design, and
    how to fix the issue. We will provide N examples of a UI screenshot
    and a set of valid design comments. Please learn how to give valid
    design comments from these examples and apply this knowledge to
    determine valid design comments for the last UI. Please be specific
    in your comments, referring to specific UI elements by their text
    label or icon, like in the examples provided. Also, please do not
    return any comments regarding user testing nor adherence to platform
    standards.
```

Text Filtering: For the provided UI and a list corresponding design comments, please filter out the incorrect design comments and return a list tuples. Each tuple contains its index i in the list, followed by True or False. The tuple would contain True if the design comment at index i in the input list is a valid design comment, and False if the design comment at index i is an invalid comment. Please analyze the UI screenshot to determine whether or not each design comment is valid. We will give N examples, where each UI screenshot is followed by a list of its corresponding design comments and an output list of tuples, where each tuple contains the list index and True/False indicating the validity of the design comment at that index. Please learn from these examples, analyzing the UI screenshot to see why each comment was considered valid or invalid. Finally, we will give a UI screenshot, followed by its corresponding design comments. Please output a list of tuples consisting of the comment's list index and an indication of each comment's validity, like in the provided examples. Please output False for the design comment if it is about consistency with the brand, user testing, or adherence to platform standards. Please only output this list of tuples and nothing else.

Bounding Box Generation: You will be providing bounding boxes coordinates for the provided UI screenshot and design comment. The bounding box will enclose a relevant region in the screenshot that is discussed in the design comment. You will use the coordinate axes along the edge of the screenshot to determine the coordinates of the bounding box. Please make sure you follow the provide coordinate axes, so that vertical bounding box coordinates are between 0 and 16 and horizontal bounding box coordinates are between 0 and 9, and format the bounding box coordinates as (left, top, right, bottom). Please do not output bounding boxes with area 0. Also, please only output the bounding box and nothing else. We will provide N examples of design comments, followed by the corresponding UI screenshot (with a coordinate axis along its edge) and a correct bounding box for the design comment in the UI screenshot based on the coordinate axis. Please learn how to determine accurate bounding boxes for the design comment in the UI screenshot based on these examples. We will provide a final design comment and UI screenshot; please apply what you have learned from the examples to determine an accurate bounding box for this final design comment and UI screenshot only.

Bounding Box Refinement: You will be refining bounding boxes for a given UI screenshot and design comment. The bounding box will enclose a relevant region in the screenshot that is discussed in the design comment. You will be given a proposed bounding box candidate and will evaluate whether or not this bounding box accurately encloses the region in the screenshot that is discussed in the comment. The proposed bounding box coordinates, in the format of (left_coordinate, top_coordinate, right_coordinate, bottom_coordinate) and is displayed as a blue box in the screenshot patch that is also provided , with some additional margin around the blue bounding box. Please reflect on whether or not this bounding box is accurate and look closely at the UI elements contained in the blue bounding box to judge its accuracy and relevance to the design comment. If the bounding box is not accurate, please output a new bounding box that you think is accurate in the format of (left_coordinate, top_coordinate, right_coordinate, bottom_coordinate), where each coordinate is determined from the coordinate axes along the edge of the UI screenshot provided earlier. Please make sure the new bounding box you output is accurate, and refer to the coordinate axes along the edge of the zoomed-in screenshot patch and the entire screenshot (provided earlier) to determine the bounding box coordinates. If the bounding box is accurate, please output 'BOUNDING BOX IS ACCURATE, PLEASE TERMINATE'. Please only output either the updated bounding or 'BOUNDING BOX IS ACCURATE, PLEASE TERMINATE' and nothing else. We will provide N examples of bounding box refinements for a given

design comment, UI screenshot, and bounding box candidate. Please
learn how to accurately refine bounding boxes for the design comment
in the UI screenshot based on these examples. We will provide a final
 design comment, UI screenshot, and bounding box candidate; please
apply what you have learned from the examples to accurately refine
the bounding box candidate for this final design comment, UI
screenshot, and the zoomed in patch showing the bounding box
candidate.

Text and Bounding Box Validation: You are given a UI screenshot, design
comment for the UI screen, and a zoomed-in patch of the UI screenshot
 showing the corresponding bounding box for the design comment.
Please evaluate the accuracy of the design comment and bounding box
with respect to the UI screenshot. The bounding box is displayed as a
 blue box in the zoomed-in screenshot patch, and is supposed to
contain the region in the UI screen that is targeted by the design
comment. Please first evaluate if the design comment is valid for the
 provided UI screenshot, i.e. if it correctly points out a design
issue and suggests an accurate way to fix it. Please analyze the
provided UI screenshot to assess the comment's validity. If the
design comment is valid, please next evaluate whether the blue box in
 zoomed-in UI screenshot contains the region that is relevant to the
design comment. If the design comment is invalid and the blue box
still contains a region in the UI screenshot with design issues,
please return the label 'Incorrect Comment'. If the comment is valid,
 but the blue box does not contain the region relevant to the comment
, please return the label 'Incorrect Bbox'. If the comment is invalid
 and the blue box does not contain a region with design issues,
please return the label 'Both Incorrect'. Finally, if the design
comment is valid and the blue box correctly contains a region in the
UI that is relevant to the comment, please return the label 'Both
Correct'. Please only return the appropiate label and nothing else.
We will give N examples, the UI screenshot (labeled 'UI Screenshot'),
 followed by the design comment (labeled 'Design Comment'), a zoomed-
in screenshot patch showing the blue bounding box (labeled 'Zoomed-in
 Patch'), and finally the correct label (labeled 'Label') for the
accuracy of the UI screenshot, design comment, and corresponding
bounding box. Please learn from these examples, to see how to
correctly categorize the design comment and its corresponding
bounding box by accuracy. Finally, we will give a UI screenshot,
design comment, and a zoomed-in patch showing the corresponding blue
bounding box. Please apply what you have learned from the examples to
 correctly classify the accuracy of the design comment and its
corresponding bounding box.

Text Refinement: You will be refining the design comment for a specific
region in a UI screenshot. You will be given a UI screenshot, a
zoomed-in patch of the screenshot with a blue box containing the
region of interest, and a design comment for the UI region inside the
 blue box. Please evaluate whether or not the design comment
accurately describes the design issue for the UI region inside the
blue box. If the design comment is accurate, please output 'COMMENT
IS ACCURATE, PLEASE TERMINATE'. If the design comment is not accurate
, please refine the design comment to the accurate and output this
accurate design comment for the region of interest, following the
same format as the input design comment. We will provide N examples
of bounding box refinements for each UI screenshot, region of
interest, and design comment candidate for the region of interest.
Please learn how to accurately refine the design comment for the
region of interest in the UI screenshot based on these examples. We
will provide a final UI screenshot, region of interest, and design
comment candidate for the region of interest; please apply what you
have learned from the examples to accurately refine design comment
candidate for this final UI screenshot and region of interest. Please

```
     only output the refined comment or 'COMMENT IS ACCURATE, PLEASE
   TERMINATE' and nothing else.
```

## A.4 QUALITATIVE ANALYSIS

### A.4.1 QUALITATIVE ANALYSIS OF OUTPUTS FROM PIPELINE, BASELINE, AND FINETUNED LLM

We qualitatively analyzed the outputs from our pipeline, baseline, and finetuned Llama-3.2 11b. Figures 5, 6, and 7 illustrate the design feedback and corresponding bounding boxes generated by our pipeline (using Gemini-1.5-pro) for a diverse set of 12 UIs. Figure 8 presents two examples where our pipeline outperformed the baseline, and Figure 9 contains two examples where the baseline performed better. To enable easier comparison between the two conditions, we used the same set of initial comments from the TextGen module, as both the pipeline and baseline begin with this module.

As shown in figures 5, 6, and 7, we found that, more often than not, the pipeline generates helpful comments with reasonably accurate bounding boxes (highlighted in green). For the baseline, we observed that it frequently generates very generic comments that would apply to any UI screen and are usually not helpful, such as suggesting that at design should be tested with users or needs to be made responsive as shown in Figure 8 (Baseline, top screenshot). These comments are usually eliminated by the pipeline (Pipeline, top screenshot). Additionally, the pipeline successfully refined incorrect comments, as shown by the red and green comments in the top screenshot, and filters out incorrect comments during the validation stages as shown in both screenshots. For bounding boxes, those generated by the pipeline are usually tighter and closer to the correct region compared to the baseline, which often generates large, unspecific bounding boxes that encompass a significant portion of the screen, as shown by the bounding boxes in Figures 8 and 9. This demonstrates the effectiveness of iterative refinement and validation in improving bounding box accuracy. Furthermore, the large bounding boxes generated by the baseline would decrease the chance of the IoU being zero, which may have inflated the average IoU shown in Tables 2 and 3.

The pipeline sometimes eliminated valid comments, as shown in both examples in Figure 9, where the green comments were accurate comments that were eliminated. In the top screenshot, the pipeline retained only one inaccurate comment, although its bounding box was significantly improved. In the bottom screenshot, the pipeline produced a less accurate bounding box around the red buttons compared to the baseline, though these instances are rare.

We found that fine-tuned Llama-3.2 generated a very limited range of comments, primarily focusing on text readability, visual clutter, and generic critiques about the need for improved visual appeal. This limited range could be due to the over-representation of such critiques in UICrit. Figure 10 presents example outputs for two screenshots, comparing them with outputs from our pipeline. The figure shows that, in addition to its limited range of critiques, the finetuned model also produces inaccurate comments. In contrast, our pipeline generates a significantly more diverse set of comments with tighter bounding boxes, though the bounding boxes are generally less accurate than those from the fine-tuned model.

Overall, the pipeline generally outperforms the baseline qualitatively, reducing the generation of invalid and generic comments and outputting bounding boxes that are tighter, more specific, and closer to the target region. Furthermore, it generates a considerably more diverse set of comments compared to finetuned Llama, though its visual grounding is less accurate.

### A.4.2 QUALITATIVE ANALYSIS OF PIPELINE OUTPUTS FOR OUT OF DOMAIN UIS

Since UICrit consists of older UIs (from 2014) Duan et al. (2024b), we evaluated the pipeline's performance to determine whether it generalizes to modern UIs and other out-of-domain UIs, such as websites, using only few-shot examples selected from UICrit. Figure 11 displays the generated feedback for four modern Android UIs (the few-shot examples from UICrit are also Android UIs) from 2024, taken from Mobbin[3]. Figure 12 presents feedback for four modern iOS UIs from 2024,

---

[3]https://mobbin.com/

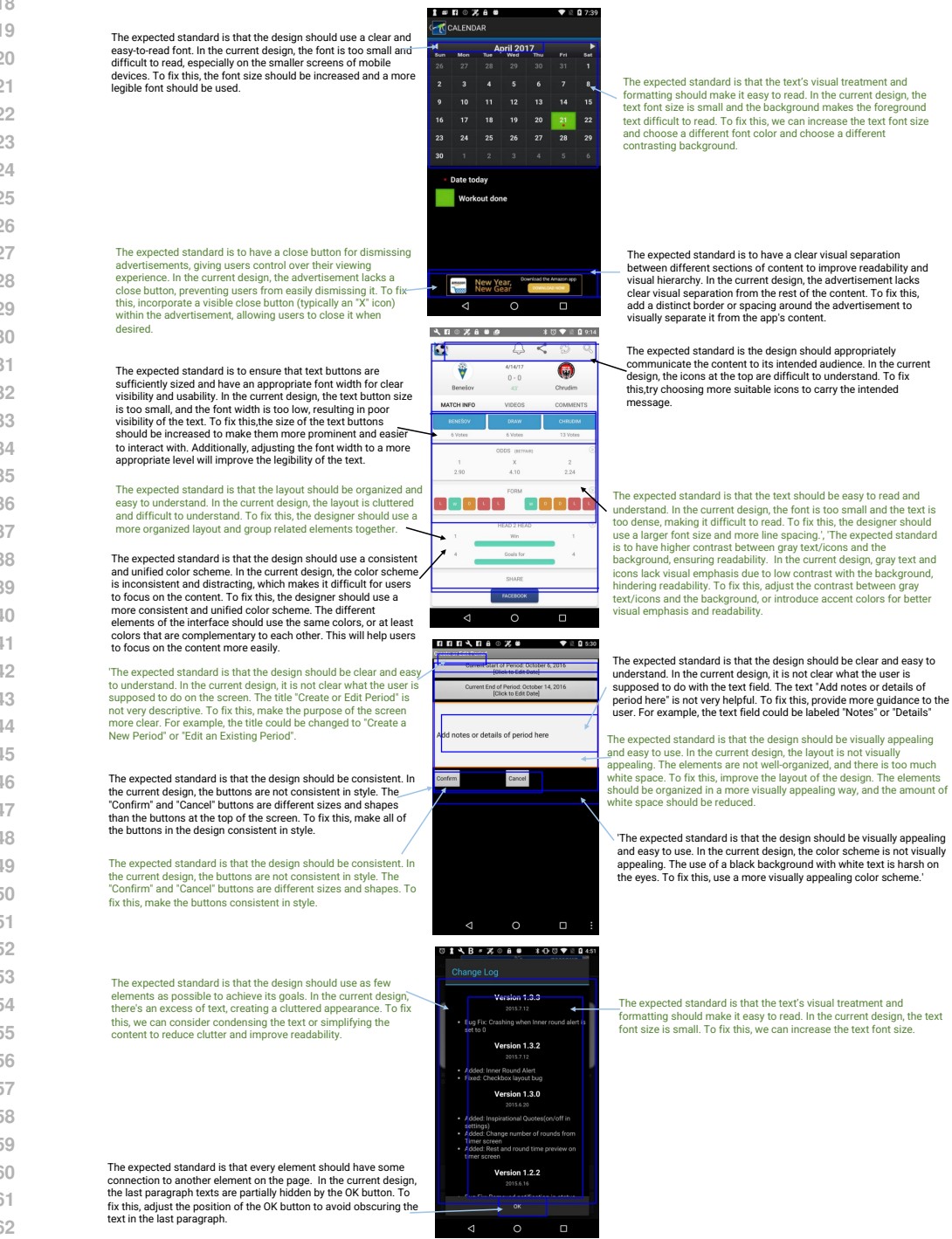

The expected standard is that the design should use a clear and easy-to-read font. In the current design, the font is too small and difficult to read, especially on the smaller screens of mobile devices. To fix this, the font size should be increased and a more legible font should be used.

The expected standard is that the text's visual treatment and formatting should make it easy to read. In the current design, the text font size is small and the background makes the foreground text difficult to read. To fix this, we can increase the text font size and choose a different font color and choose a different contrasting background.

The expected standard is to have a close button for dismissing advertisements, giving users control over their viewing experience. In the current design, the advertisement lacks a close button, preventing users from easily dismissing it. To fix this, incorporate a visible close button (typically an "X" icon) within the advertisement, allowing users to close it when desired.

The expected standard is to have a clear visual separation between different sections of content to improve readability and visual hierarchy. In the current design, the advertisement lacks clear visual separation from the rest of the content. To fix this, add a distinct border or spacing around the advertisement to visually separate it from the app's content.

The expected standard is to ensure that text buttons are sufficiently sized and have an appropriate font width for clear visibility and usability. In the current design, the text button size is too small, and the font width is too low, resulting in poor visibility of the text. To fix this, the size of the text buttons should be increased to make them more prominent and easier to interact with. Additionally, adjusting the font width to a more appropriate level will improve the legibility of the text.

The expected standard is the design should appropriately communicate the content to its intended audience. In the current design, the icons at the top are difficult to understand. To fix this, try choosing more suitable icons to carry the intended message.

The expected standard is that the layout should be organized and easy to understand. In the current design, the layout is cluttered and difficult to understand. To fix this, the designer should use a more organized layout and group related elements together.

The expected standard is that the text should be easy to read and understand. In the current design, the font is too small and the text is too dense, making it difficult to read. To fix this, the designer should use a larger font size and more line spacing.', 'The expected standard is to have higher contrast between gray text/icons and the background, ensuring readability. In the current design, gray text and icons lack visual emphasis due to low contrast with the background, hindering readability. To fix this, adjust the contrast between gray text/icons and the background, or introduce accent colors for better visual emphasis and readability.

The expected standard is that the design should use a consistent and unified color scheme. In the current design, the color scheme is inconsistent and distracting, which makes it difficult for users to focus on the content. To fix this, the designer should use a more consistent and unified color scheme. The different elements of the interface should use the same colors, or at least colors that are complementary to each other. This will help users to focus on the content more easily.

'The expected standard is that the design should be clear and easy to understand. In the current design, it is not clear what the user is supposed to do on the screen. The title "Create or Edit Period" is not very descriptive. To fix this, make the purpose of the screen more clear. For example, the title could be changed to "Create a New Period" or "Edit an Existing Period".

The expected standard is that the design should be clear and easy to understand. In the current design, it is not clear what the user is supposed to do with the text field. The text "Add notes or details of period here" is not very helpful. To fix this, provide more guidance to the user. For example, the text field could be labeled "Notes" or "Details"

The expected standard is that the design should be consistent. In the current design, the buttons are not consistent in style. The "Confirm" and "Cancel" buttons are different sizes and shapes than the buttons at the top of the screen. To fix this, make all of the buttons in the design consistent in style.

The expected standard is that the design should be visually appealing and easy to use. In the current design, the layout is not visually appealing. The elements are not well-organized, and there is too much white space. To fix this, improve the layout of the design. The elements should be organized in a more visually appealing way, and the amount of white space should be reduced.

The expected standard is that the design should be consistent. In the current design, the buttons are not consistent in style. The "Confirm" and "Cancel" buttons are different sizes and shapes. To fix this, make the buttons consistent in style.

'The expected standard is that the design should be visually appealing and easy to use. In the current design, the color scheme is not visually appealing. The use of a black background with white text is harsh on the eyes. To fix this, use a more visually appealing color scheme.'

The expected standard is that the design should use as few elements as possible to achieve its goals. In the current design, there's an excess of text, creating a cluttered appearance. To fix this, we can consider condensing the text or simplifying the content to reduce clutter and improve readability.

The expected standard is that the text's visual treatment and formatting should make it easy to read. In the current design, the text font size is small. To fix this, we can increase the text font size.

The expected standard is that every element should have some connection to another element on the page. In the current design, the last paragraph texts are partially hidden by the OK button. To fix this, adjust the position of the OK button to avoid obscuring the text in the last paragraph.

Figure 5: Illustration of four example outputs from the pipeline. The screenshots are marked with the output bounding boxes, and the generated comments are shown, each pointing to its corresponding bounding box. Helpful comments with reasonably accurate bounding boxes are highlighted in screen.

sourced from DesignVault[4], and Figure 13 illustrates feedback for five modern websites from 2024,

---

[4] https://designvault.io/

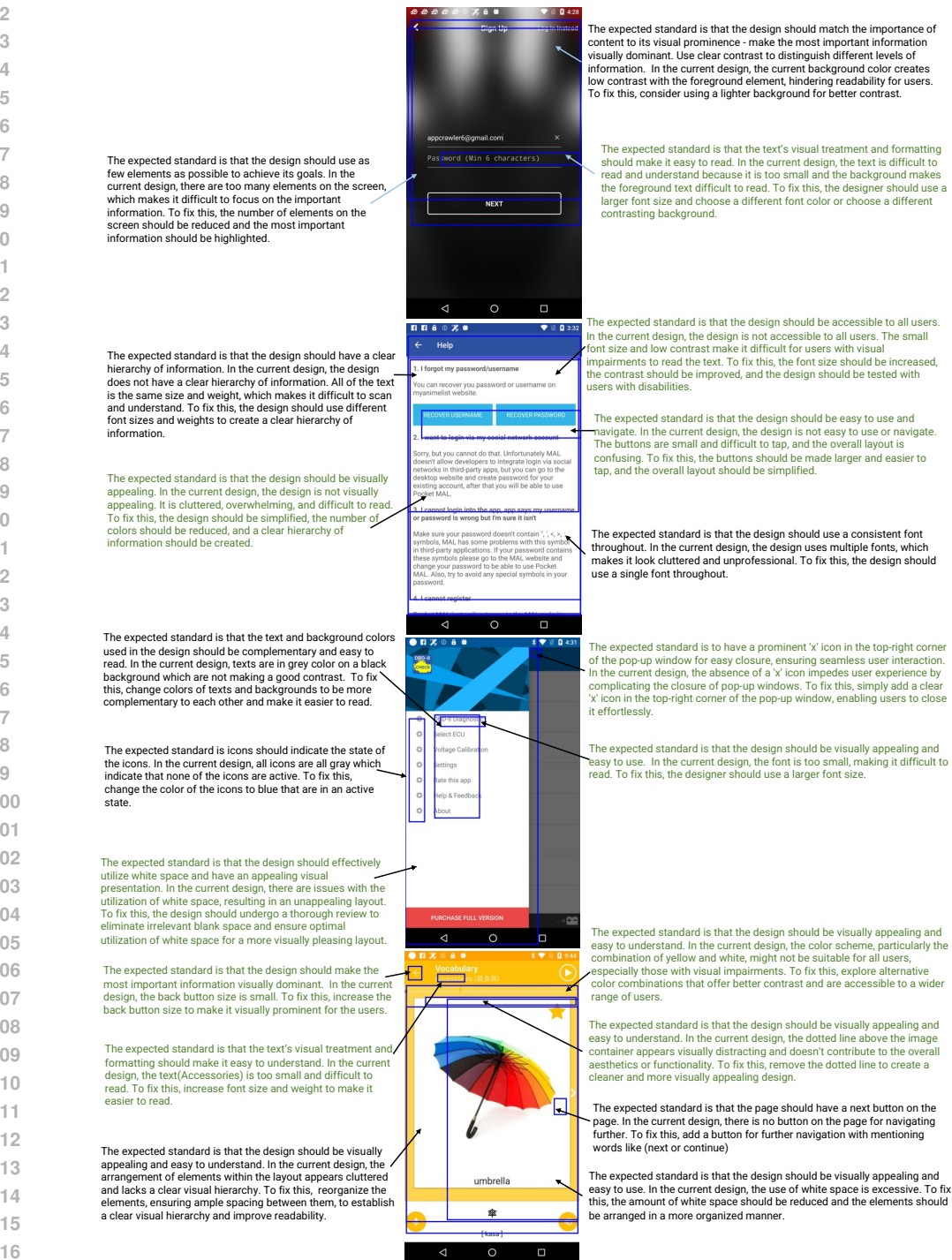

Figure 6: Illustration of four example outputs from the pipeline. The screenshots are marked with the output bounding boxes, and the generated comments are shown, each pointing to its corresponding bounding box. Helpful comments with reasonably accurate bounding boxes are highlighted in screen.

also taken from Mobbin. In these figures, helpful comments with reasonably accurate bounding boxes are highlighted in green.

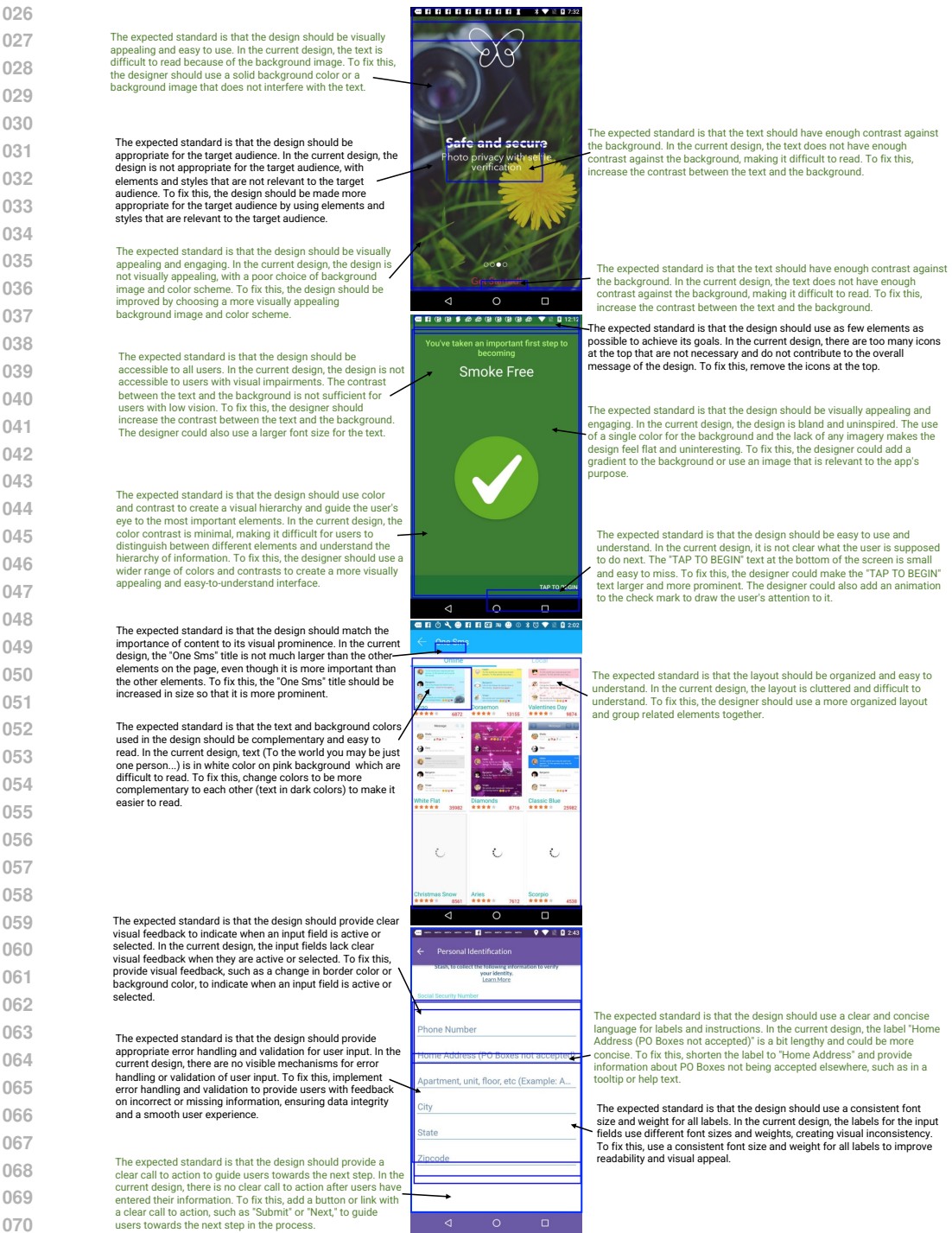

Figure 7: Illustration of four example outputs from the pipeline. The screenshots are marked with the output bounding boxes, and the generated comments are shown, each pointing to its corresponding bounding box. Helpful comments with reasonably accurate bounding boxes are highlighted in screen.

The pipeline was able to provide helpful feedback with reasonably accurate bounding boxes for these out-of-domain UIs. It performed surprisingly well on the modern iOS UIs, with results comparable to those for the UIs from the test split of UICrit, as shown in Figures 5, 6, 7, and 8. Additionally, the pipeline even managed to generate helpful feedback and bounding boxes for websites. While design

principles often overlap between mobile and web interfaces, their layouts and screenshot dimensions differ significantly. This suggests that the LLM was able to generalize and adapt its knowledge to generate and refine bounding boxes for website screenshots, despite only being trained with few-shot examples from mobile screenshots, which are very different.

An interesting observation is that, since websites have more screen space, they are generally more complex and information-dense than mobile UIs (Gazzawe, 2017). We found one instance where the pipeline incorrectly flagged a relatively simple website as being too complex (i.e. having too many elements) in Figure 13 (second screen from the bottom), likely because it evaluated the complexity based on the mobile standards presented in the few-shot examples. However, the pipeline did correctly critique the bottom screenshot in the same figure for being overly complex, showing that it can appropriately identify this issue in some cases.

### A.4.3 REFINING BOUNDING BOXES

While the bounding boxes could be improved qualitatively, as shown in Figures 5, 6, 7, and 8, there are straightforward approaches to easily improve the bounding box accuracy. For instance, the DOM tree representation of the UI contains the exact bounding boxes of UI elements and element groups. This information could be used to refine the output bounding boxes for the elements/groups discussed in the critiques by finding the closest bounding box from the DOM tree via IoU comparison, distances between the bounding box centers and sizes, or utilizing an LLM for matching, as was done in Zhao et al. (2024). This DOM representation is available through the UI's XML code, or the internal UI mockup representation available in design tools like Figma. In the case that the DOM tree is not available, we could use a screen object parser (Wu et al., 2021) to extract UI element and group locations from the screenshot.

We demonstrate the results of using the DOM tree (taken from the XML-based Android View Hierarchy available in RICO(Deka et al., 2017)) to refine the pipeline's bounding boxes for some of the UICrit UIs in Figures 5, 6, 7, and 8. As discussed above, we matched the generated bounding boxes with those from the UI elements and groups in the DOM tree via an IoU threshold. The results are shown in Figures 14 and 15 and illustrate that this simple refinement method significantly improves the bounding boxes. This step could potentially be applied at the end of the pipeline to clean up the generated bounding boxes.

### A.5 ANALYSIS OF ITERATIVE REFINEMENT

Figure 16 illustrates an example of iterative bounding box refinement (conditioned on the comment) by BoxRefine, which terminates on a significantly more accurate bounding box. Figure 17 illustrates an example of comment refinement (conditioned on the bounding box) by TextRefine, which terminates on an accurate comment on the poor layout of the region inside the bounding box.

We calculated the average number of bounding box refinements, which were 1.25 for Gemini-1.5-pro and 0.88 for GPT-4o, as well as the average number of comment refinements, which were 1.48 for Gemini-1.5-pro and 1.17 for GPT-4o. Additionally, we estimated the expected number of LLM calls required for a complete run of the pipeline, including the small fraction sent for further refinement by Validation. The expected number of calls is 7.16 for Gemini-1.5-pro and 6.70 for GPT-4o.

### A.6 HUMAN EVALUATION METHOD

Figure 18 shows a snippet of the form used by human design experts to rate the quality of individual comments and rank the comment sets for the three different conditions.

Given the limited availability of UI design experts and the extensive evaluation required per UI screen for a detailed comparison across the three conditions, only the Gemini-1.5-pro outputs for 33 UI screenshots were rated. To better represent the UI design space in this sample, we maximized the diversity of the UI screenshots by randomly sampling an even number of UIs from each of the UI task categories identified by Duan et al. (2024b). We followed their method of clustering by task descriptions from UICrit to obtain the task clusters. These 33 UIs were split into 6 groups for rating, with three participants assigned to each group. The rating and ranking study took approximately 1 hour. We recruited 18 design experts for this study. Five of the participants had 2-4 years of

design experience, and the rest had 6-10 years. Their areas of design expertise include mobile, web, interaction, and user experience research.

### A.7 ALGORITHMS FOR GENERATING FEW-SHOT EXAMPLES FOR BOUNDING BOX REFINEMENT

Algorithm 1 details the steps for generating the few-shot refinement examples for a selected bounding box. The few-shot generation algorithm entails perturbing the bounding box coordinates by decreasing amounts and adding the perturbations to the list of few-shot examples. The algorithm for perturbing a bounding box is also shown in Algorithm 1.

---
**Algorithm 1** Generate Bounding Box Refinement Few-shot Examples

---
**Require:** the bounding box to be perturbed $input\_bbox$, the fraction that the bounding box's coordinates will be perturbed $perturb\_frac$

**Ensure:** The coordinates of $input\_bbox$ perturbed by $perturb\_frac$

1: **function** GENERATE_PERTURB($input\_bbox$, $perturb\_frac$)
2:     Compute $left\_margin$, $right\_margin$, $top\_margin$, $bottom\_margin$
3:     $all\_perturbed \leftarrow []$
4:     **for** $x\_perturb$ in $[-perturb\_frac \times left\_margin, perturb\_frac \times right\_margin]$ **do**
5:         **for** $y\_perturb$ in $[-perturb\_frac \times top\_margin, perturb\_frac \times bottom\_margin]$ **do**
6:             Update bounding box location based on $x\_perturb$, $y\_perturb$
7:             Add perturbed bounding box to $all\_perturbed$
8:         **end for**
9:     **end for**
10:    $final\_perturbed \leftarrow []$
11:    Compute $width$ and $height$ of the input bounding box
12:    **for** each $perturbed\_bbox$ in $all\_perturbed$ **do**
13:        **for** $width\_fraction$ in $[-perturb\_frac, perturb\_frac]$ **do**
14:           **for** $height\_fraction$ in $[-perturb\_frac, perturb\_frac]$ **do**
15:              Update bounding box size based on $width\_fraction$ and $height\_fraction$
16:           **end for**
17:        **end for**
18:    **end for**
19:    $filtered\_perturbed \leftarrow remove\_invalid\_perturbed\_bbox(final\_perturbed, input\_bbox)$
20:    $final\_bbox \leftarrow random.choice(filtered\_perturbed)$
21:    **return** $final\_bbox$
22: **end function**

---

**Require:** Bounding box $bbox$, maximum number of perturbations of $bbox$ in the list of fewshot refinement examples $max\_num\_perturb$

**Ensure:** A list of bounding boxes coordinates that are perturbed versions of $bbox$ in decreasing amounts, where $bbox$ is the last item in the list.

23: **function** GENERATE_PERTURBED_FEWSHOT_EXAMPLES($bbox$, $max\_num\_perturb$)
24:    $perturb\_options \leftarrow$ LIST(range($max\_num\_perturb + 1$))
25:    $num\_perturb \leftarrow$ RANDOM_CHOICE($perturb\_options$)
26:    $perturb\_list \leftarrow []$
27:    **for** $j \leftarrow num\_perturb$ **to** 1 **do**
28:       $perturb\_frac \leftarrow j/max\_num\_perturb$
29:       $output\_bbox \leftarrow$ GENERATE_PERTURB($bbox$, $perturb\_frac$)
30:       $perturb\_list$.append($output\_bbox$)
31:    **end for**
32:    $perturb\_list$.append($bbox$)
33:    **return** $perturb\_list$
34: **end function**

---

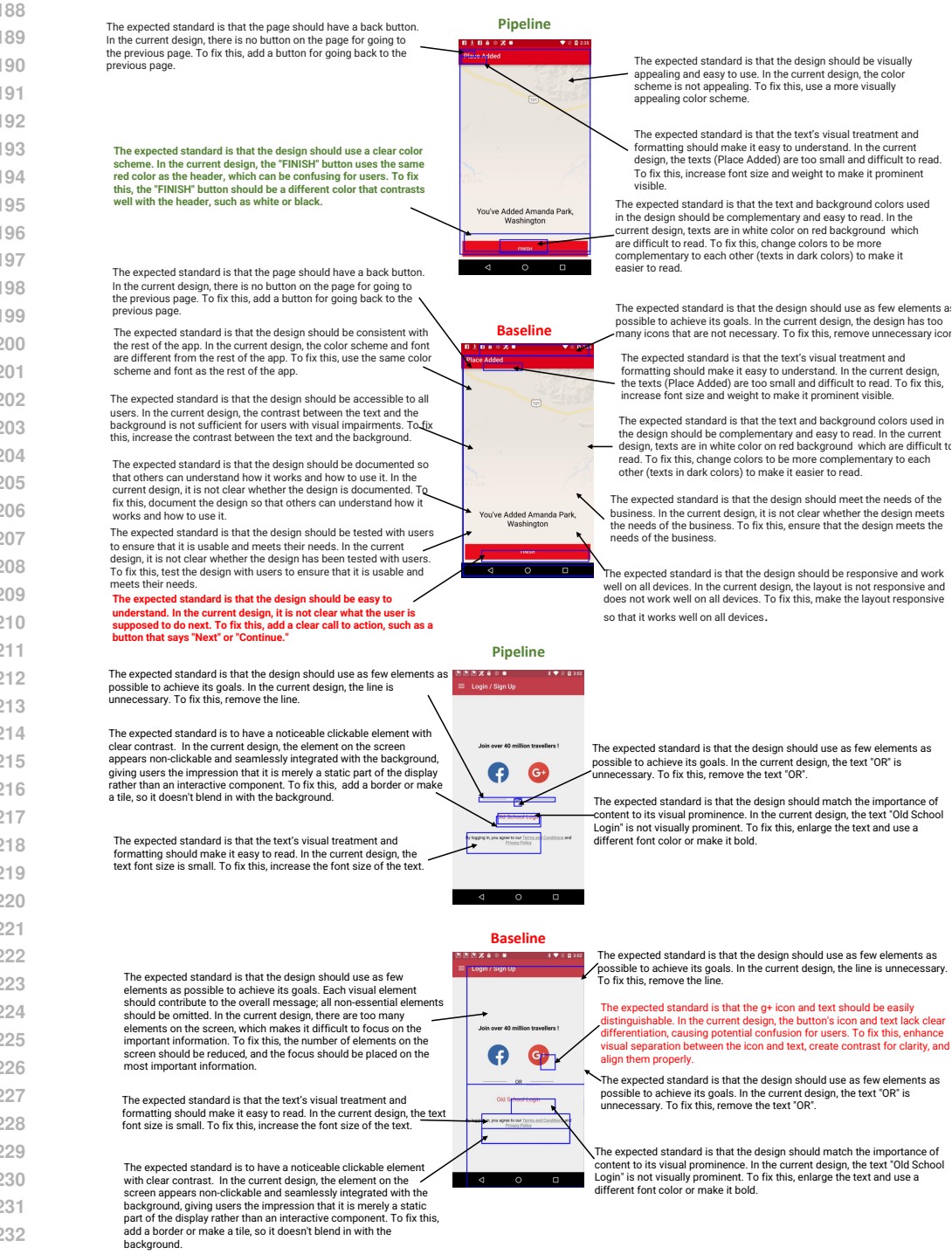

Figure 8: Illustration of outputs from the pipeline and baseline, highlighting two cases where our pipeline outperformed the baseline. The screenshots are marked with the output bounding boxes, and the generated comments are shown, each pointing to its corresponding bounding box (some comments have the same bounding box). Both the pipeline and baseline begin with the TextGen module, so we used the same initial comments from TextGen for both conditions for easier comparison. In the top example, the pipeline produced more accurate bounding boxes, eliminated several generic and unhelpful comments, and refined an inaccurate comment (red) into a more accurate one (green). In the bottom example, the pipeline produced more considerably more accurate bounding boxes, and eliminated an invalid comment (red).

Figure 9: Illustration of outputs from the pipeline and baseline, highlighting two cases where the baseline outperformed our pipeline. The screenshots are marked with the output bounding boxes, and the generated comments are shown, each pointing to its corresponding bounding box (some comments have the same bounding box). Both the pipeline and baseline begin with the TextGen module, so we used the same initial comments from TextGen for both conditions for easier comparison. For the top example, while a lot of the comments from the baseline are inaccurate, the pipeline eliminated the only correct comment (green) and only kept an invalid comment (red), though its bounding box is considerably more accurate. In the bottom example, the pipeline removed two valid comments (green) and some invalid ones, and also made the bounding box around the comment regarding the red buttons less accurate.

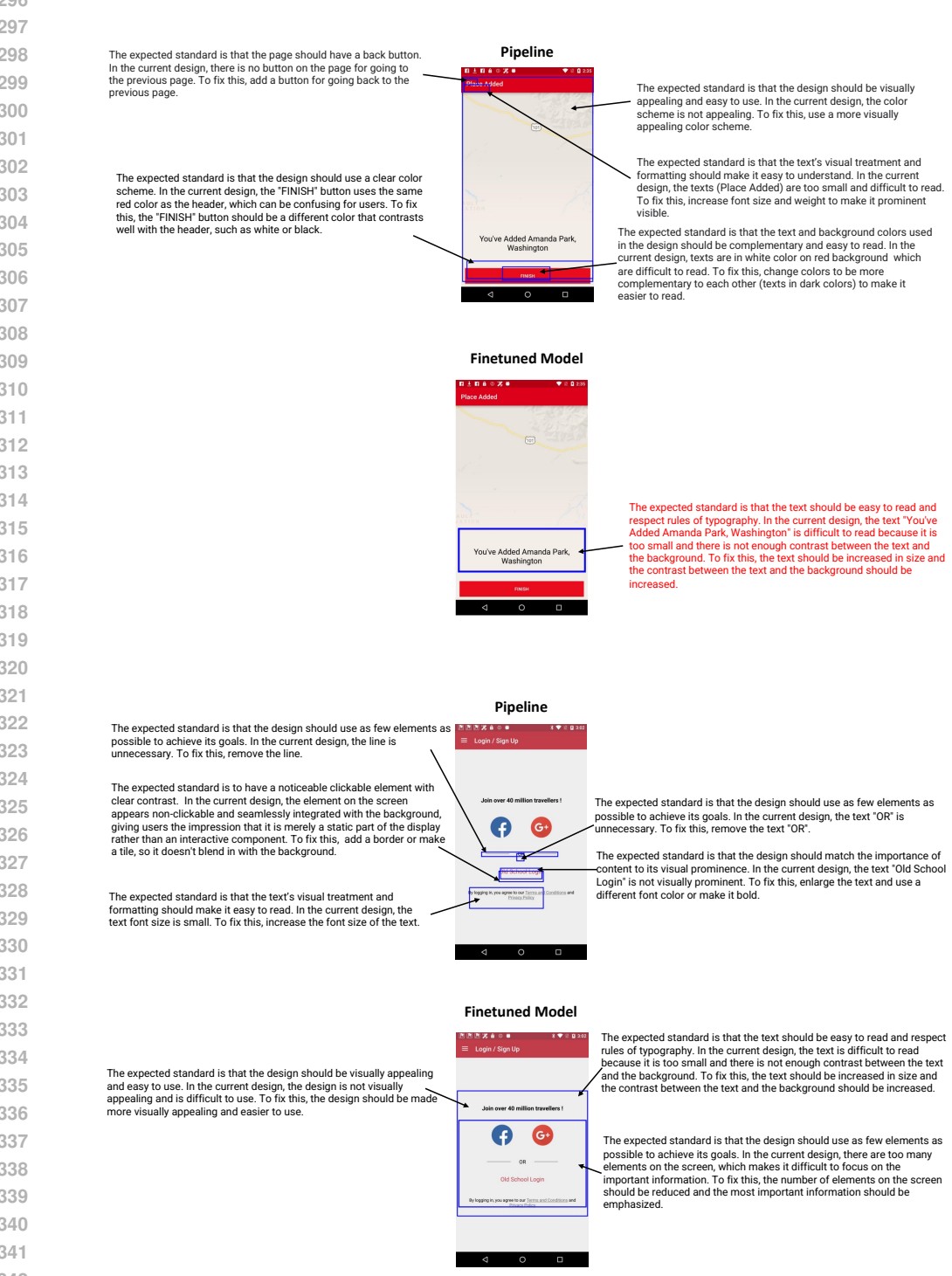

Figure 10: Illustration of outputs from the pipeline and finetuned Llama-3.2 11b. The screenshots are marked with the output bounding boxes, and the generated comments are shown, each pointing to its corresponding bounding box (some comments have the same bounding box). The fine-tuned model produces a limited range of critiques, some of which are inaccurate (red), though the bounding boxes are generally accurate. In contrast, the pipeline generates a significantly more diverse set of critiques, and its bounding boxes are tighter but generally less accurate.

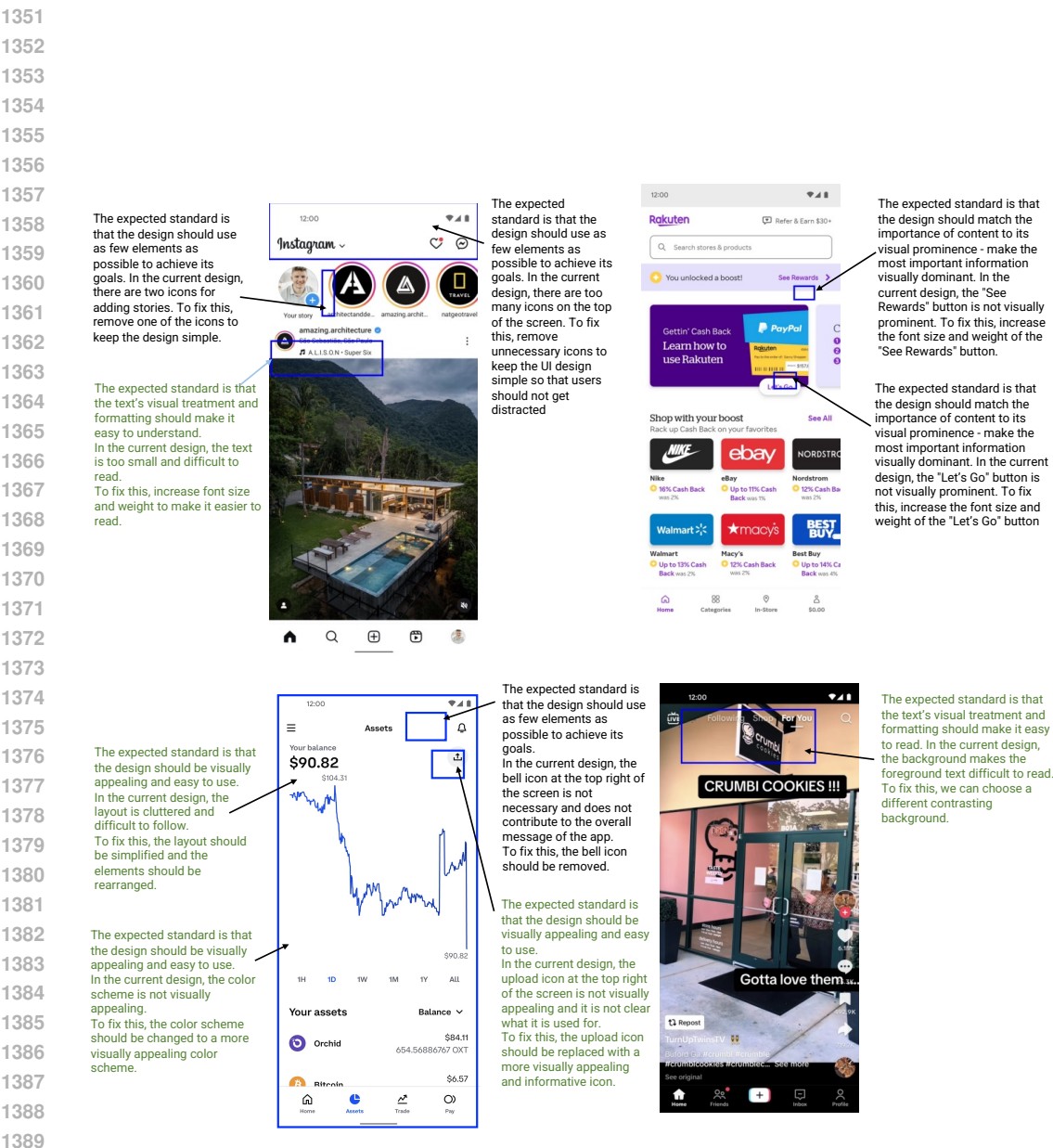

Figure 11: Example design feedback and bounding boxes generated by our pipeline for four modern Android UIs (from 2024). These UIs are out-of-domain inputs, as we used fewshot examples from only UICrit, which consists of older UIs (from 2014). The screenshots are marked with the output bounding boxes, and the generated comments are shown, each pointing to its corresponding bounding box. Helpful comments with reasonably accurate bounding boxes are highlighted in screen.

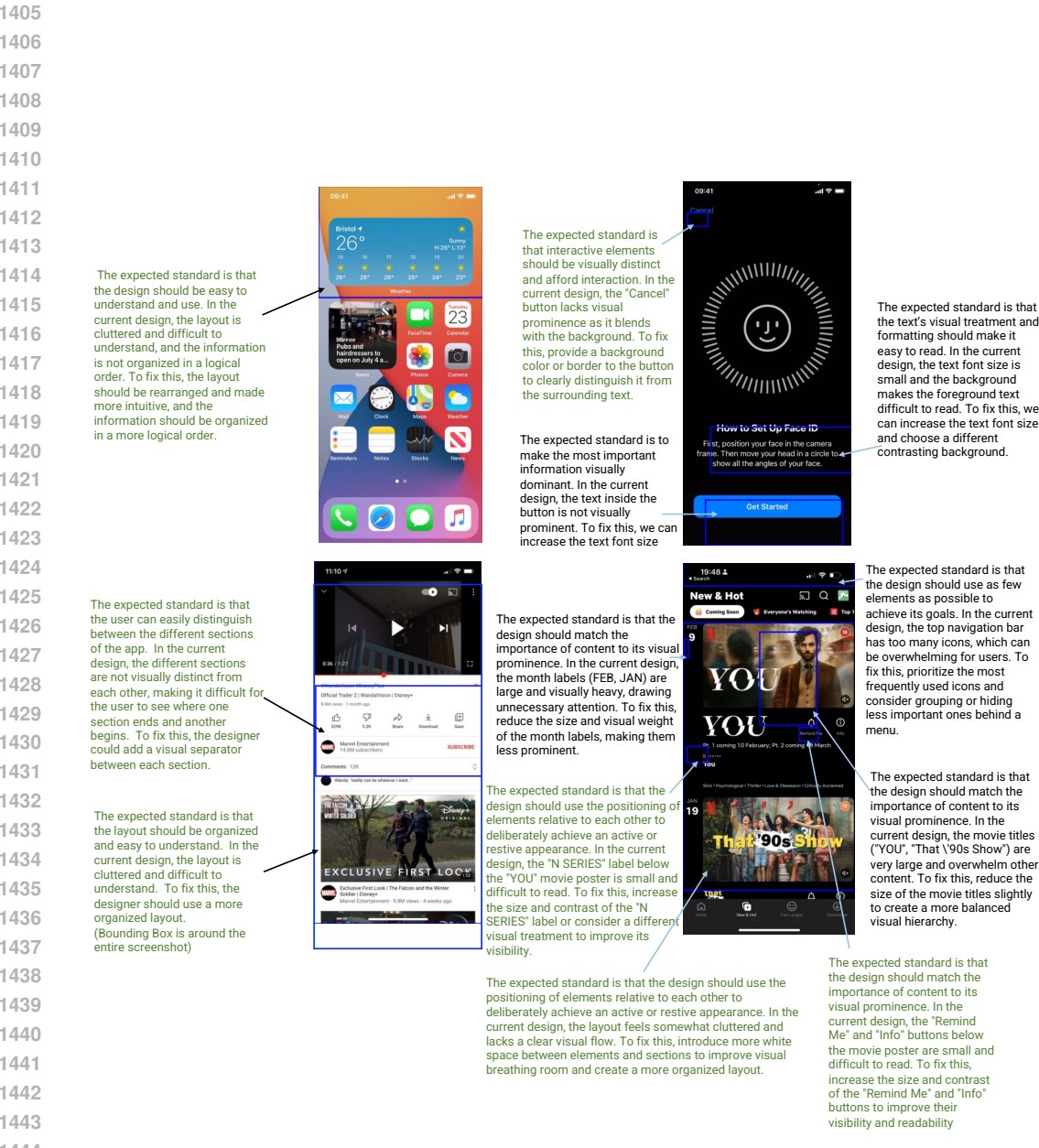

Figure 12: Example design feedback and bounding boxes generated by our pipeline for four modern iOS UIs (from 2024). These UIs are out-of-domain inputs, as we used fewshot examples from only UICrit, which consists of older UIs (from 2014). The screenshots are marked with the output bounding boxes, and the generated comments are shown, each pointing to its corresponding bounding box. Helpful comments with reasonably accurate bounding boxes are highlighted in screen.

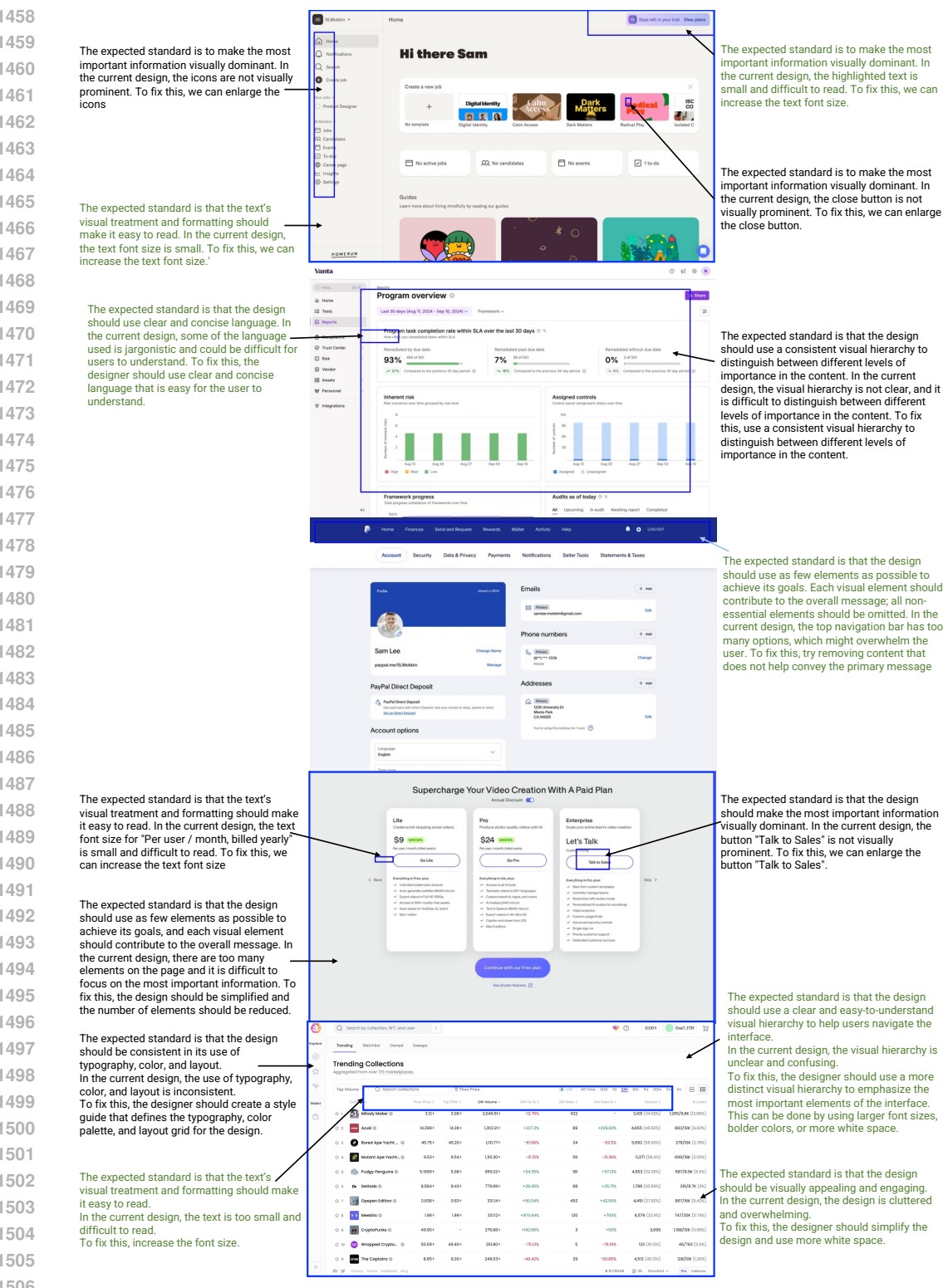

Figure 13: Example design feedback and bounding boxes generated by our pipeline for five modern websites (from 2024). These websites are out-of-domain inputs, as we used fewshot examples from only UICrit, which consists of older mobile UIs (from 2014) that differ significantly from modern websites. The screenshots are marked with the output bounding boxes, and the generated comments are shown, each pointing to its corresponding bounding box. Helpful comments with reasonably accurate bounding boxes are highlighted in screen.

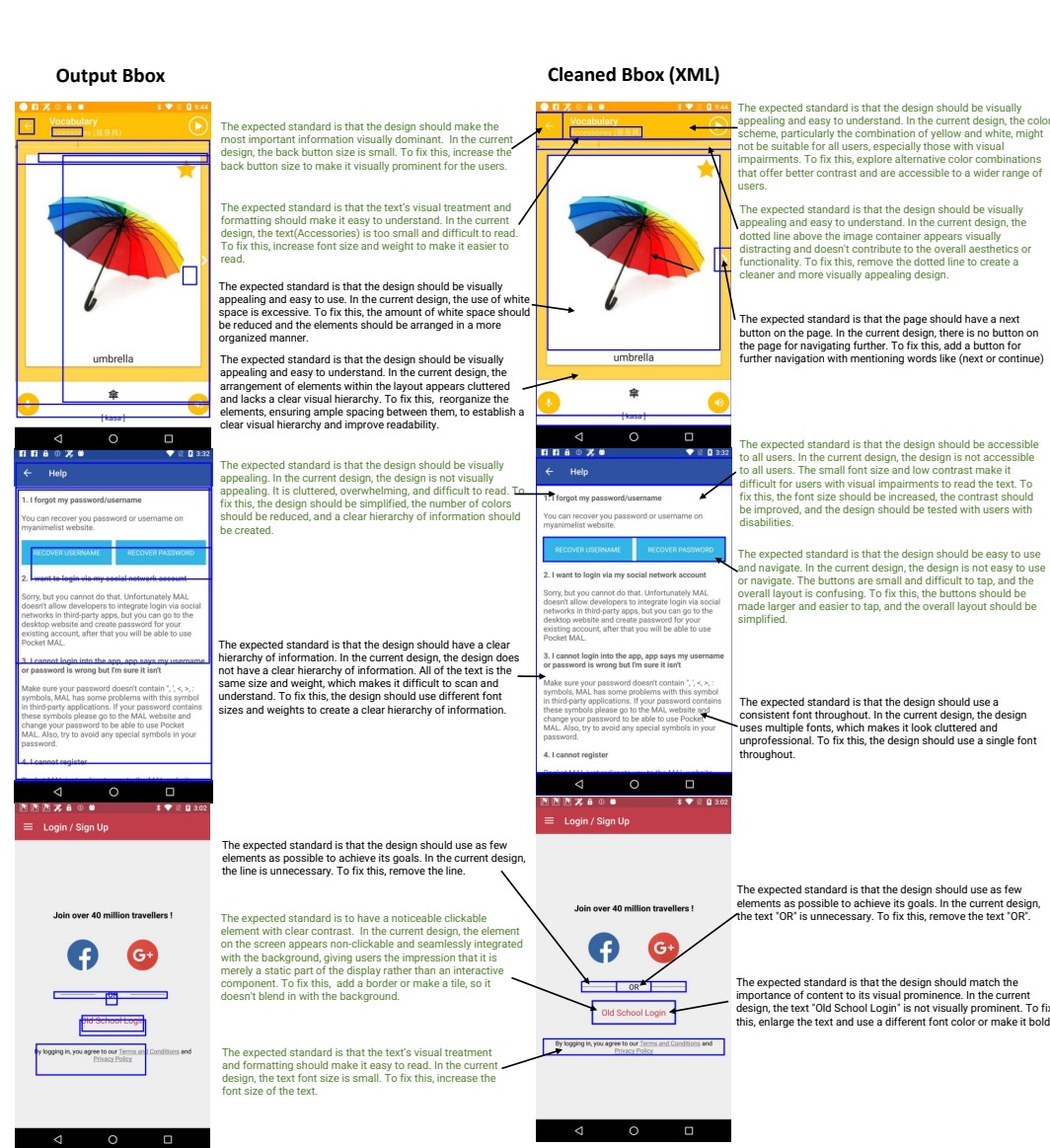

Figure 14: Side by side comparison of the bounding boxes generated by the pipeline ("Output Bbox") and the output bounding boxes after refinement using a simple method that locates the nearest elements and groups from the DOM tree based on an IoU threshold ("Cleaned Bbox (XML)"). This refinement approach significantly improves the quality of the generated bounding boxes.

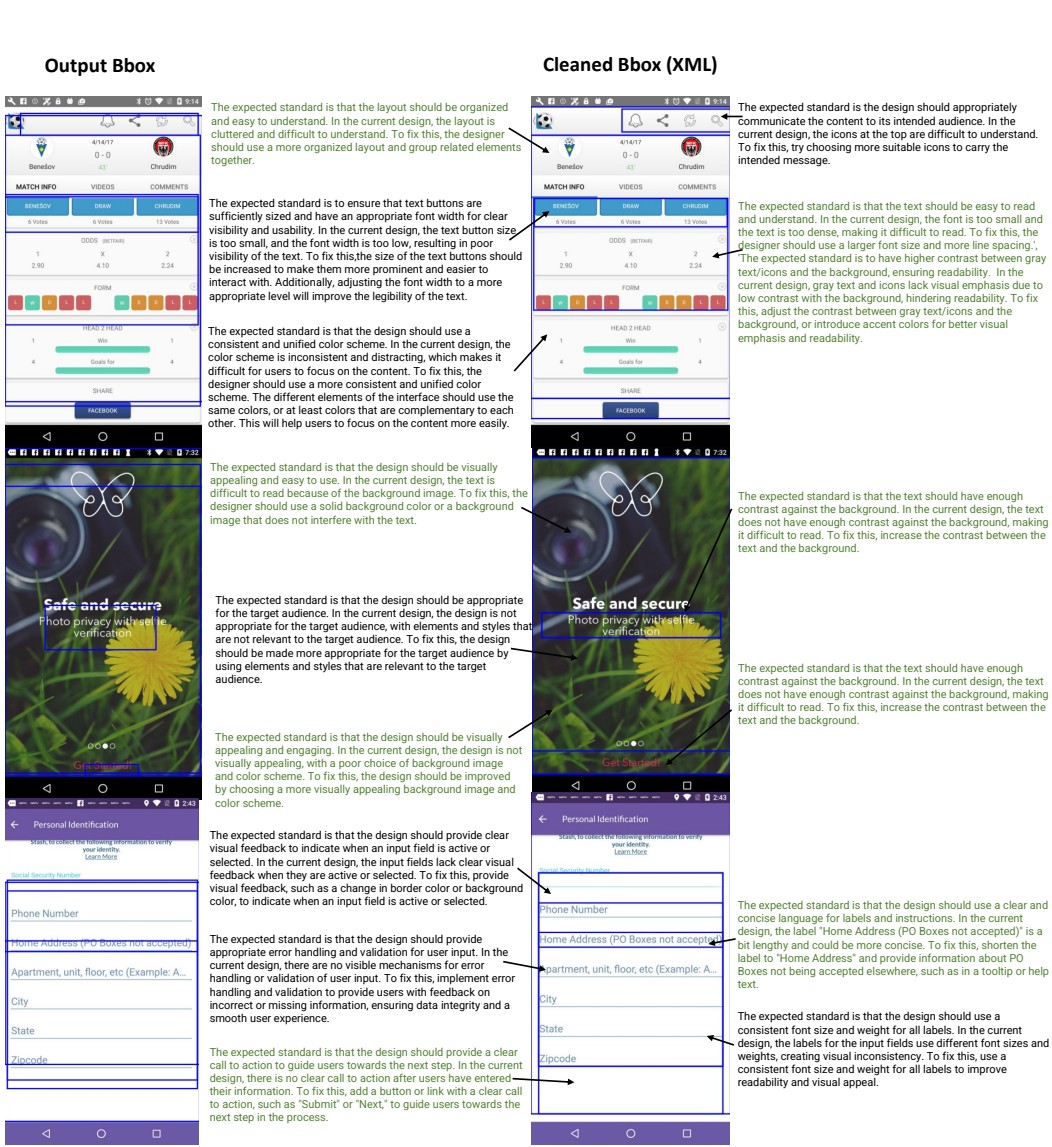

Figure 15: Side by side comparison of the bounding boxes generated by the pipeline ("Output Bbox") and the output bounding boxes after refinement using a simple method that locates the nearest elements and groups from the DOM tree based on an IoU threshold ("Cleaned Bbox (XML)"). This refinement approach significantly improves the quality of the generated bounding boxes.

Start · End

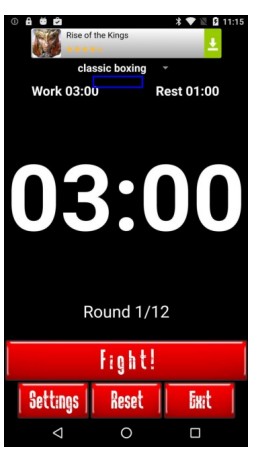 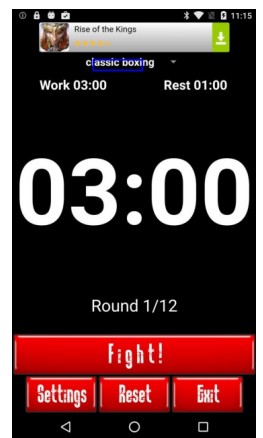

**Comment:** The expected standard is that the design should match the importance of content to its visual prominence. In the current design, the text "classic boxing" is not visually prominent. To fix this, we can increase the font size of the text "classic boxing".

Figure 16: An example of iterative bounding box refinement, with the comment it is conditioned on displayed on the right. The bounding box in the first screenshot ('Start') is the output from BoxGen. The refinement process progressively improves the bounding box, terminating on a significantly more accurate bounding box ('End').

Start · End · Screenshot

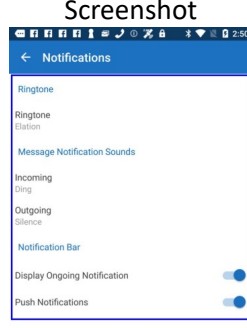 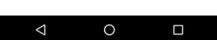

The expected standard is that design should convey a clear message In the current design, it does not provide enough information to the users to understand what the app itself is all about. To fix this, redesign it by adding additional information with features to communicate the content to its intended users.

The expected standard is that the design should be consistent throughout the app. In the current design, the "Ringtone" section and the "Message Notification Sounds" section are not consistent with each other. The "Ringtone" section has a dropdown menu, while the "Message Notification Sounds" section does not. To fix this, the designer should make the two sections consistent with each other. For example, both sections could have dropdown menus.

The expected standard is that the design should be easy to understand and use. In the current design, the layout of the notification settings is confusing and difficult to follow. To fix this, the designer should reorganize the layout to make it more intuitive.

Figure 17: An example of iterative comment refinement, with the bounding box it is conditioned on displayed on the right. The first comment ('Start') was classified as incorrect by the Validation but has an accurate corresponding bounding box. The refinement process progressively improves the comment, terminating with an accurate comment on the poor layout of the region in the bounding box. ('End').

**Comment Set Ranking**

Set A (All Comments)

1. The expected standard is that the text's visual treatment and formatting should make it easy to read. In the current design, the text font size is small and the background makes the foreground text difficult to read. To fix this, we can increase the text font size and choose a different contrasting background.

Set B (All Comments)

1. The expected standard is to have high contrast and a visually appealing background that complements the design's overall aesthetic. In the current design, the black background lacks visual appeal. To fix this, consider a lighter background or a textured black option to improve contrast and visual interest.

Set C (All Comments)

1. The expected standard is that the design should use as few elements as possible to achieve its goals. Each visual element should contribute to the overall message; all non-essential elements should be omitted. In the current design, there are too many elements on the screen, making it difficult to focus on any one thing. To fix this, the designer should remove any unnecessary elements from the screen.

Please rank each set of comments, as a whole, based on their overall quality. Please rank them in decreasing quality.

|       | 1 | 2 | 3 |
|-------|---|---|---|
| Set A | ○ | ○ | ○ |
| Set B | ○ | ○ | ○ |
| Set C | ○ | ○ | ○ |

**Individual Comment Rating**

Set C Comment (10 of 15) *

The expected standard is that the design should be visually appealing and easy to use. In the current design, the text is not aligned properly. To fix this, the designer should align the text to the center.

○ Invalid

○ Partially Valid

○ Valid

Figure 18: The form used for individual comment quality rating and comment set ranking.

