# OpenReview forum: "Visual Prompting with Iterative Refinement for Design Critique Generation"
_ICLR.cc/2025/Conference — Submitted to ICLR 2025_

### Official Review · Reviewer_S1K2 · 2024-10-25

**Soundness:** 3
**Presentation:** 3
**Contribution:** 2
**Rating:** 6
**Confidence:** 3

**Summary:**

This work focuses on generating high-quality design critiques, where the inputs are an UI screen and design guidelines and outputs are design comments along with corresponding boxes that map each comment to specific region in the screenshot. It proposes an iterative visual prompting approach for UI critique, where LLMs are leveraged to iteratively refine both the text output and bounding boxes using few-shot samples tailored for each step.

**Strengths:**

1.	It focuses on a novel, practical and difficult task - generating design critiques, which will have great impact on UI design by saving much design effort and accelerating design process.

2.	The insight into designing each component of the prompting pipeline is clearly explained.

3.	The experiments demonstrate the proposed method effectively.

**Weaknesses:**

1.	There is no qualitative results. Without good examples, I cannot concretely know what design critiques can be generated. For example, are they diverse enough? Can the difficult design critiques be generated? Without bad examples, I cannot clearly know the shortcomings of the proposed method.

2.	There is no discussion about the cost in terms of time or number of calls. The prompting pipelines involves many rounds of refinement and relies on the visual capabilities of LLMs. All of these will make it a slow one, which will influence the user experience.

3.	There is no comparison with methods based on fine-tuning, e.g., fine-tuning open-source multimodality LLMs like LLAVA or continually fine-tuning the one from Bravo et al. (2023). If fine-tuning based methods perform comparable or even better than the proposed prompting pipeline, we should use fine-tuning since the prompting pipeline with iterative refinement is slow and expensive.

======================

After discussion with authors, most of my concerns have been addressed. While the techniques may not be highly novel and could have a limited impact on the broader AI/ML community, I appreciate its contribution to the specific domain of AI-assisted design. I would like to raise my score from 5 to 6.

**Questions:**

How many rounds of iterative refinement are used in the experiment (averagely)? What will happen if we use less rounds of refinement or allow for more rounds of refinement?

---

> ### Author Response · Authors · 2024-11-19
>
> We thank Reviewer S1K2 for your insightful comments. We have revised the paper based on your feedback and uploaded the revised draft. We addressed your questions and concerns as follows:
>
> > Weakness 1: There is no qualitative results. Without good examples, I cannot concretely know what design critiques can be generated. For example, are they diverse enough? Can the difficult design critiques be generated? Without bad examples, I cannot clearly know the shortcomings of the proposed method.
>
> We added Section A.4 in the Appendix that contains the results of a qualitative analysis. This section contains Figure 5, which presents two example outputs that demonstrate how our pipeline improves upon the baseline. Figure 6 (Appendix) includes two examples where our pipeline performed worse than the baseline. We have also updated Section 5.4 with a summary of these qualitative results. Please let us know if you want further clarification and examples.
>
> > Questions: How many rounds of iterative refinement are used in the experiment (averagely)? What will happen if we use less rounds of refinement or allow for more rounds of refinement?
>
> > Weakness 2: There is no discussion about the cost in terms of time or number of calls. The prompting pipelines involves many rounds of refinement and relies on the visual capabilities of LLMs. All of these will make it a slow one, which will influence the user experience.
>
> We computed the average number of refinement steps for both bounding boxes (1.25 steps for Gemini and 0.88 for GPT-4o) and design comments (1.48 steps for Gemini and 1.17 for GPT-4o). If we reduce the rounds of bounding box refinement, the resulting IoU would likely approach that of BBox Generation (Table 2), and more comment + bounding box pairs may be removed during Validation. A similar effect is expected with fewer rounds of comment refinement.
>
> We set a high limit of 10 refinement steps, and the iterative refinement rarely reaches this limit before deciding that the bounding box or comment is accurate. Therefore, allowing for additional rounds of refinement would likely not significantly affect the outcome. We added Section A.5 in the Appendix, which discusses and illustrates iterative refinement outputs, and also includes a brief cost analysis of the average number of refinement steps and the expected number of LLM calls for a full pipeline run.
>
> > Weakness 3: There is no comparison with methods based on fine-tuning, e.g., fine-tuning open-source multimodality LLMs like LLAVA or continually fine-tuning the one from Bravo et al. (2023). If fine-tuning based methods perform comparable or even better than the proposed prompting pipeline, we should use fine-tuning since the prompting pipeline with iterative refinement is slow and expensive.
>
> Thanks for bringing up the general question of fine-tuning versus prompting. We want to emphasize that the focus of our paper is to investigate how far LLMs can go on critique tasks using prompting alone, which is a research question that has merits on its own. That said, we agree with you that fine-tuning might have advantages, especially when a more sizable dataset is available. Based on the UICrit dataset, we conducted additional experiments to address your question, and these details have been added in the revision. Namely, we have finetuned Llama 3.2 11b on UICrit for 3 epochs and have obtained the following results:
>
> | Metric | Finetuned Llama-3.2 11b | Gemini-1.5-pro | GPT-4o |
> | - | - | - | - |
> | Comment Similarity | 0.842 |  0.702  | 0.701 |
> | Estimated IoU | 0.230 | 0.199 | 0.275 |
>
> While finetuned Llama 3.2 has comparable Estimated IoU to our pipeline and higher comment similarity, qualitatively, it generates a very limited set of comments that cover text readability, visual clutter, and general comments on visual appeal. In contrast, our pipeline produces a more diverse range of critiques. Therefore, the fine-tuned model could not adequately replace the pipeline.
>
> We have updated Table 2 with these quantitative results. We have added Figure 7 in the Appendix, which provides example outputs from both the fine-tuned Llama model and our pipeline to illustrate this, and we added the results of this qualitative analysis to Section A.4 of the Appendix. We have also updated Section 5.4 and the Discussion (Section 7) with this new finding.
>
>
> Please let us know if you have any further questions or would like to see additional changes.

---

> > ### Comment · Reviewer_S1K2 · 2024-11-20
> >
> > Thanks for your feedback. Most of my concerns have been addressed. While the techniques may not be highly novel and could have a limited impact on the broader AI/ML community, I appreciate its contribution to the specific domain of AI-assisted design. I would like to raise my score from 5 to 6.

---

> > > ### Author Response · Authors · 2024-11-20
> > > **Thank you!**
> > >
> > > Thank you very much for your response and for raising your score. We truly appreciate you taking the time to carefully read our rebuttal and acknowledge the value of our contribution to AI-assisted design. We understand your point about the novelty and broader impact. We plan to further explore the applicability of our appraoch in other domains in our future work.

---

> > > > ### Author Response · Authors · 2024-11-23
> > > > **Review score**
> > > >
> > > > Dear Reviewer S1K2,
> > > >
> > > > It seems that the new score is not reflected in OpenReview yet. Would you please check on this? Thanks again for your constructive feedback!

---

### Official Review · Reviewer_WLXH · 2024-11-02

**Soundness:** 2
**Presentation:** 3
**Contribution:** 2
**Rating:** 3
**Confidence:** 4

**Summary:**

The paper presents an LLM based method to provide a text based critique (design feedback) on a UI design, given an image of that design.  In addition, the method can ground its feedback using bounding boxes overlaid on that image to indicate the basis for the feedback.

The technical contribution is the engineering of an LLM based pipeline, which consists of LLMs coupled together in multiple stages: 1) TextGen – generates design comments as text given a task/system prompt without any grounding; 2) TextFilter – a further LLM is used to prune spurious comments; 3) BoxGen – creates the grounding bounding boxes for each comment; 4) BoxRefine – Improves accuracy of the bounding boxes.

The paper explores the efficacy of various LLMs (GPT4 vs. Gemini1.5) at the task and ablates these stages and parameter choices within them to justify the pipeline design.  The experiments are done on Duan et al’s UICrit dataset and baselined against the recent Duan et al. CHI 2024 paper which addresses the same task.

**Strengths:**

Automated UI design critique seems a well motivated task; there is a potential for UX design efficiencies in using an LLM for this purpose although without actually sequencing / simulating the interaction with an interface the feedback is limited only to superficial judgements around appearances of UI elements.  There may be value therefore in this idea as a kind of visual ‘linter’ for UI design that could be an assistant or validation check for less experienced designers.  That said, the paper is not the first to propose the task – see the recent (2024) dataset and baseline of Duan et al. cited.

The paper is clearly written – the stages of the LLM pipeline are straightforward and reproducabile, as relevant system prompt examples are given in the appendix.

Given that the paper essentially proposes a 4 stage pipeline, the necessary ablation is in place to show the value of each of the 4 stages.  A brief baseline comparison is made to the prior work of Duan et al (CHI 2024) on the same dataset (UICrit).

**Weaknesses:**

There is limited technical innovation or scientific insight in the paper.  Essentially, the authors report a way to couple together several LLM based processes to create a grounded description of the image in the context of UX design guidelines (derived from the classic Jakub HCI paper on the same).  Whilst the design justifications for each stage are given, there is no real insight into the LLM’s capabilities or limitations.  Why is an LLM able to do this?  Which guidelines can it advise best on, and why?  Rather the innovation is presented at face value, as an engineering result.

The paper introduces a formalised notation for the UI design critique tasks which seems unused later in the paper.  Whilst it is welcome to have a clear task definition, I do not see the value of the math notation defined in Section 3 when it is not actually used later i.e. in the Method section of the paper.  The use of this formalism appears to distract from what is otherwise largely an engineering based contribution sequencing multiple LLMs together.

An attempt is made to assess the alignment of the UI critiques with human critiquers.  However the experiment is performed on only 33 UI examples, using 18 humans.  Whilst it is laudable to try to quantify human alignment, picking just 33 UIs out of a the huge design space of UIs can give no meaningful information on the performance of the LLMs at the general UI design critique task.  Also the task differs from the method proposed in the  baseline Duan et al. baseline, in that users are asked only to validate the comments for each UI region (i.e. check correctness) they are not asked to comment on the entire UI (i.e. check completeness).

I’m confused as to the purpose of Section 6 which seems to apply the UI critiquing 4 stage pipeline to the general open-world object detection/description task similar to Grounding DINO.  Why would this task be relevant to the proposed pipeline?  This entire section should be removed in my view.

Minor: The paper describes the pipeline as having 3 stages (comprising 6 LLMs) but then describes 4 stages (TextGen, TextFilter, BoxGen, BoxRefine).  pp.9 ‘Section’ typo in cross-reference:

**Questions:**

Overall this paper addresses the relatively novel task of UI design critique using LLMs.  However the contribution is largely engineering based, creating a particular sequence of LLM promptings to create UI design critiques.  There is limited depth of insight as to the capabilities and limitations of this pipeline, beyond a baseline comparison to one prior work  and some ablation of the design itself.  The user study is so sparse as to not really provide any insight, and deviates without good justification from the approach of the baseline method.  An extra study (Section 6) that seems out of context, as well as some math formulation (Section 3) serve to bulk up the paper but don’t provide useful additionality to the exposition or problem defined.

---

> ### Author Response · Authors · 2024-11-19
>
> We thank Reviewer WLXH for your insightful comments. We have revised the paper based on your feedback and uploaded the revised draft. We addressed your questions and concerns as follows:
>
> > Weakness 1: There is limited technical innovation or scientific insight in the paper. Essentially, the authors report a way to couple together several LLM based processes to create a grounded description of the image in the context of UX design guidelines (derived from the classic Jakub HCI paper on the same). Whilst the design justifications for each stage are given, there is no real insight into the LLM’s capabilities or limitations. Why is an LLM able to do this? Which guidelines can it advise best on, and why? Rather the innovation is presented at face value, as an engineering result.
>
> The cited paper by Duan et. al. (CHI 2024) has studied the capabilities and limitations of LLMs for design critique, justifying that LLMs have the potential to do this task based on their generalization capabilities to evaluate arbitrary UIs, reasoning abilities, and amenability to text-based design guidelines. They also found that LLMs advised the best on visual guidelines because visual guidelines contain more straightforward checks for aspects like color contrast and alignment. Since these topics have already been investigated by Duan et. al. (CHI 2024), we build upon these findings. For instance, our approach provides tailored fewshot examples to help general purpose LLMs better capture the nuances of design evaluation and address their limitations identified by Duan et al. (CHI 2024). We want to point out that our work is not just an engineering contribution. Our work contributes scientific insights in that it involves designing and investigating what works and what doesn’t by using LLMs as building blocks for a complex multimodal task. Additionally, we demonstrated the potential of this approach to be applicable and useful for other tasks.
>
>
> This paper pushes the boundaries of what prompting alone can achieve for complex multimodal tasks. We provide insights into prompting for such tasks, specifically whether prompting can improve upon existing baselines for the complex multimodal task of design critique, how to implement it effectively, and whether the performance improvements from our method generalize to other multimodal tasks. For technical contributions, we introduce visual grounding with iterative refinement as a general technique applicable to various multimodal tasks and designed few-shot examples in a generalizable way that enhances performance. Our investigation focuses on the potential of LLMs using prompting alone, given that fine-tuning is highly resource-intensive in terms of data and computational requirements, which highlights the value of exploring lightweight prompting approaches. Without this work, there would be limited guidance on setting up these tasks with prompting alone to maximize performance.
>
> > Weakness 2: The paper introduces a formalised notation for the UI design critique tasks which seems unused later in the paper. Whilst it is welcome to have a clear task definition, I do not see the value of the math notation defined in Section 3 when it is not actually used later i.e. in the Method section of the paper. The use of this formalism appears to distract from what is otherwise largely an engineering based contribution sequencing multiple LLMs together.
>
> Thank you for pointing out that the formalized notation for the UI design critique task is unnecessary. We also agree, and have removed the mathematical notation from Section 3.

---

> > ### Author Response · Authors · 2024-11-19
> >
> > (Continued from Above Comment)
> > > Weakness 3: An attempt is made to assess the alignment of the UI critiques with human critiquers. However the experiment is performed on only 33 UI examples, using 18 humans. Whilst it is laudable to try to quantify human alignment, picking just 33 UIs out of a the huge design space of UIs can give no meaningful information on the performance of the LLMs at the general UI design critique task. Also the task differs from the method proposed in the baseline Duan et al. baseline, in that users are asked only to validate the comments for each UI region (i.e. check correctness) they are not asked to comment on the entire UI (i.e. check completeness).
> >
> > We acknowledge that 33 UI examples is a relatively small sample size. However, we were constrained by the limited availability of design experts and their time, and the need for a thorough comparison across the three conditions, as noted in lines 911-913. To better represent the UI design space in this sample, we maximized the diversity of the UIs by randomly sampling an even number of UIs from each of the Task categories shown in Figure 5 of Duan et. al. (UIST 2024). We followed their method of clustering by Task Descriptions from UICrit to obtain the task clusters. We added this detail to Section A.6 of the Appendix, where we moved the details of the human validation study method. Another thing to note is that this set of 33 UIs is significantly larger than the set of 6 UIs that were used in the human validation study from the baseline paper (Duan et. al. (UIST 2024)).
> >
> > The human validation study from the baseline paper (Duan et. al. (UIST 2024)) also only validated the comments and bounding box accuracy and did not check for completeness by asking participants to provide additional comments that were missed. The only difference between our method and theirs is that, in their study, participants accounted for both the generated comment and generated bounding box in their ratings. In contrast, we adopted a more rigorous approach to estimate bounding box accuracy: multiple authors manually determined and agreed upon the ground truth bounding box for each output comment, which was then presented to participants alongside the comment. Participants rated only the comment accuracy, while the IoU of the output bounding boxes was computed against the established ground truth. We have further clarified these details in lines 421-426.
> > > Weakness 4: I’m confused as to the purpose of Section 6 which seems to apply the UI critiquing 4 stage pipeline to the general open-world object detection/description task similar to Grounding DINO. Why would this task be relevant to the proposed pipeline? This entire section should be removed in my view.
> >
> > The purpose of Section 6 is to demonstrate that our pipeline can generalize to other grounded multimodal tasks, such as open-world object detection and description. Since this pipeline could theoretically support any grounded multimodal task, we wanted to assess whether its performance improvements over the baseline would extend beyond design critique. Showing that the pipeline is effective across multiple tasks highlights its broader applicability, making it a more significant contribution than if it were effective for design critique alone.
> >
> > > Weakness Minor: The paper describes the pipeline as having 3 stages (comprising 6 LLMs) but then describes 4 stages (TextGen, TextFilter, BoxGen, BoxRefine). pp.9 ‘Section’ typo in cross-reference:
> >
> > The descriptions in Section 4 are for the 6 individual LLMs (TextGen, TextFilter, BoxGen, BoxRefine, Validation, and TextRefine) and not the 3 stages. We discussed this in lines 198-200
> >
> > We also have fixed the Section reference typo – thank you for catching it!
> >
> > > Questions: Overall this paper addresses the relatively novel task of UI design critique using LLMs. However the contribution is largely engineering based, creating a particular sequence of LLM promptings to create UI design critiques. There is limited depth of insight as to the capabilities and limitations of this pipeline, beyond a baseline comparison to one prior work and some ablation of the design itself. The user study is so sparse as to not really provide any insight, and deviates without good justification from the approach of the baseline method. An extra study (Section 6) that seems out of context, as well as some math formulation (Section 3) serve to bulk up the paper but don’t provide useful additionality to the exposition or problem defined.
> >
> > We have individually addressed each of these questions in our responses above to their corresponding weakness points.
> >
> > Please let us know if you have any further questions or would like to see additional changes.

---

> > > ### Author Response · Authors · 2024-11-23
> > > **Follow up**
> > >
> > > Dear Reviewer WLXH,
> > >
> > > We are wondering if our rebuttal has addressed your concerns. Would you please get back to us on our responses?
> > >
> > > Thank you!

---

> > > > ### Comment · Reviewer_WLXH · 2024-11-24
> > > >
> > > > Whilst the rebuttal/revision addresses some (but not all) of the presentation and exposition issues raised in my review, my fundamental concerns about the experimental setup, rigor and the novelty of this work are discussed but not fixed. Therefore I will keep my original score.

---

> ### Author Response · Authors · 2024-11-27
>
> Dear Reviewer WLXH,
>
> Thank you very much for your responses. We are glad that our rebuttal and revision have addressed some of your concerns. Here we would like to further address your concerns regarding the experimental setup/rigor and the novelty. We sincerely hope you can reconsider your score given the progress we have made in addressing your points, and the additional information we provide here.
>
> ### Re: the experimental setup and rigor
> We understand your concern about the scale of the experiments. However, we hope you can evaluate the scale of our experiments in the context of multimodal critique generation, which is very complex due to its subjective and open-ended nature and requires participants with design expertise. Compared to previous studies on this topic reported in the literature, our study with human design expert ratings is significantly more comprehensive and rigorous than other studies involving human design experts. Unlike crowdworkers, human professional design experts are rare and are hence, challenging to recruit. Despite this, we conducted a study with 18 design experts who collectively evaluated 33 UIs, with each UI's comments rated by three designers to ensure validity. In comparison, other studies involving human design experts typically include only 5–7 experts [1, 2, 3, 4] or 14 experts [5], and evaluate considerably fewer UIs (6–17). Thus, we sincerely hope Reviewer WLXH takes into account these unique challenges we face in human evaluation for this task.
>
>
> ### Re: novelty
> Our work is the first to bring iterative refinement to the multimodal domain. There are many considerations and experimentations with trial-and-error to find the right design for the set of prompting techniques in our approach. Critique generation is a complex multimodal task, and our technique clearly outperformed the baselines. Our findings are valuable for other researchers to continue investigating the topic, and our approach has also shown promise in solving multimodal tasks beyond critique generation. We believe simplicity is the strength of our approach as it makes the work easily reproducible by others, and applicable to other problems, while still incorporating many details and careful designs in our approach. We sincerely hope you can share our view.
>
>
> References:
>
> [1] Peitong Duan, Chin-Yi Cheng, Gang Li, Bjoern Hartmann, and Yang Li. 2024. UICrit: Enhancing Automated Design Evaluation with a UI Critique Dataset. In Proceedings of the 37th Annual ACM Symposium on User Interface Software and Technology (UIST '24). Association for Computing Machinery, New York, NY, USA, Article 46, 1–17. https://doi.org/10.1145/3654777.3676381
>
> [2] Ziming Wu, Yulun Jiang, Yiding Liu, and Xiaojuan Ma. 2020. Predicting and Diagnosing User Engagement with Mobile UI Animation via a Data-Driven Approach. In Proceedings of the 2020 CHI Conference on Human Factors in Computing Systems (CHI '20). Association for Computing Machinery, New York, NY, USA, 1–13. https://doi.org/10.1145/3313831.3376324
>
> [3] Amanda Swearngin and Yang Li. 2019. Modeling Mobile Interface Tappability Using Crowdsourcing and Deep Learning. In Proceedings of the 2019 CHI Conference on Human Factors in Computing Systems (CHI '19). Association for Computing Machinery, New York, NY, USA, Paper 75, 1–11. https://doi.org/10.1145/3290605.3300305
>
> [4] Camilo Fosco, Vincent Casser, Amish Kumar Bedi, Peter O'Donovan, Aaron Hertzmann, and Zoya Bylinskii. 2020. Predicting Visual Importance Across Graphic Design Types. In Proceedings of the 33rd Annual ACM Symposium on User Interface Software and Technology (UIST '20). Association for Computing Machinery, New York, NY, USA, 249–260. https://doi.org/10.1145/3379337.3415825
>
> [5] Eldon Schoop, Xin Zhou, Gang Li, Zhourong Chen, Bjoern Hartmann, and Yang Li. 2022. Predicting and Explaining Mobile UI Tappability with Vision Modeling and Saliency Analysis. In Proceedings of the 2022 CHI Conference on Human Factors in Computing Systems (CHI '22). Association for Computing Machinery, New York, NY, USA, Article 36, 1–21. https://doi.org/10.1145/3491102.3517497

---

### Official Review · Reviewer_PCXV · 2024-11-02

**Soundness:** 2
**Presentation:** 3
**Contribution:** 2
**Rating:** 5
**Confidence:** 3

**Summary:**

The paper presents a method that coordinates multiple LMMs for design critique generation. The method takes as input a UI design image along with a set of design principles, and produces text comments on the design issues as well as a set of bounding boxes for visual grounding of the comments (i.e., localizing the problematic regions).

The main contribution is to improve (Duan et al., 2024b) through iterative refinement of outputs and a set of specially designed prompting techniques.

**Strengths:**

1. Building computational methods for automatic generation of design critiques is an important problem to solve.

 2. The system is well designed and illustrated, and evaluated extensively.

**Weaknesses:**

1. The scale of technical novelty is limited. What the paper actually does is to bring an existing idea, i.e., iterative output refinement for LMMs (Madaan et al., 2023; Xu et al., 2024a) into an existing problem domain, i.e., multimodal design critique generation (Duan et al., 2024b). The paper introduces several prompting techniques, such as including zoom-in image regions for the predicted bounding boxes into the visual prompts for bounding box refinement. However, the amount of novelty involved in these simple prompting methods is not significant enough. To increase the level of technical novelty, one possibility is to come up with more sophisticated prompting techniques, e.g., to generate better few-shot examples than those created via simple random perturbation for BoxRefine, Validation and TextRefine, or to iteratively modify the UI image beyond adding coordinate markers for more efficient bounding box refinement.

2. The quantitative scores of (Duan et al., 2024b) in terms of Comment Similarity and Estimated IoU (used in Table 2) are missing, and should be added.

3. The improvement upon (Duan et al., 2024b) in terms of comment quality (from 0.45 to 0.47 in Table 3) is small. For an incremental work, a more noticeable performance boost is expected, e.g., from 0.45 to 0.5 that almost lies midway between (Duan et al., 2024b) and Human.

**Questions:**

In the human evaluation, why were the participants not asked to rate bounding box accuracy as in (Duan et al., 2024b)? Is “BBox IoU” in Table 3 computed by comparing with the ground truth bounding boxes? If so, is it possible for “BBox IoU” to penalize a predicted bounding box that is valid but different from any ground truth ones?

---

> ### Author Response · Authors · 2024-11-19
>
> We thank Reviewer PCXV for your insightful comments.  We have revised the paper based on your feedback and uploaded the revised draft. We addressed your questions and concerns as follows:
>
> > Weakness 1: The scale of technical novelty is limited. What the paper actually does is to bring an existing idea, i.e., iterative output refinement for LMMs (Madaan et al., 2023; Xu et al., 2024a) into an existing problem domain, i.e., multimodal design critique generation (Duan et al., 2024b). The paper introduces several prompting techniques, such as including zoom-in image regions for the predicted bounding boxes into the visual prompts for bounding box refinement. However, the amount of novelty involved in these simple prompting methods is not significant enough. To increase the level of technical novelty, one possibility is to come up with more sophisticated prompting techniques, e.g., to generate better few-shot examples than those created via simple random perturbation for BoxRefine, Validation and TextRefine, or to iteratively modify the UI image beyond adding coordinate markers for more efficient bounding box refinement.
>
> We acknowledge that the iterative refinement concept has been previously applied in various text-only domains (e.g., Madaan et al., 2023; Xu et al., 2024a). However, our work introduces novel adaptations of iterative refinement tailored to the multimodal design critique domain, which has unique challenges such as critique-specific visual grounding and refinement, and spatially contextual critique refinement. We also found that our method generalizes to other multimodal tasks beyond design critique.
>
> Although our methods seem simple, which we feel it is a strength instead of a weakness; it took extensive investigation and iteration to design these methods. We considered other designs for prompting and fewshot examples, but found that our current setup had the best performance. For instance, we tried another few-shot design for bounding box refinement. This technique involves selecting the initial bounding box location based on visual similarity of the region it contained in the fewshot UI to that of the region contained by the input bounding box proposal of the input screenshot. This bounding box is then gradually moved closer to the ground truth bounding box for the fewshot UI to simulate the refinement process. However, we found that the simpler approach of randomly perturbing the bounding box actually gave better results (IoU 0.357 (random perturbation) vs 0.333 (visual similarity match)). We have updated Section A.2.1 in the Appendix with a description of this alternative few shot prompting method and a comparison of its performance with random perturbation (lines 702-708).
>
>
> This paper pushes the boundaries of what prompting alone can achieve for complex multimodal tasks. We provide insights into prompting for such tasks, specifically whether prompting can improve upon existing baselines for the complex multimodal task of design critique, how to implement it effectively, and whether the performance improvements from our method generalize to other multimodal tasks. We feel these constitute valuable contributions for others to explore the topic further.
>
> > Weakness 2: The quantitative scores of (Duan et al., 2024b) in terms of Comment Similarity and Estimated IoU (used in Table 2) are missing, and should be added.
>
> Thank you for pointing this out. We have updated Table 2 with the metrics from the baseline.

---

> ### Author Response · Authors · 2024-11-19
>
> (Continued from Above Comment)
> > Weakness 3: The improvement upon (Duan et al., 2024b) in terms of comment quality (from 0.45 to 0.47 in Table 3) is small. For an incremental work, a more noticeable performance boost is expected, e.g., from 0.45 to 0.5 that almost lies midway between (Duan et al., 2024b) and Human.
>
> While it is true that the performance boost for individual comment quality is less than expected, the comment set quality ranking is midway between (Duan et al., 2024b) and Human ((Duan et al., 2024b): 2.3, Pipeline: 2.0, Human: 1.7). The overall quality of the comment set is arguably more important than the average quality of each individual comment, as the feedback for a UI screen is typically presented as a set of comments. Also, participants were instructed to consider more aspects when ranking the comment sets, including the overall quality of the comments and comprehensiveness of the set.
>
> > Questions: In the human evaluation, why were the participants not asked to rate bounding box accuracy as in (Duan et al., 2024b)? Is “BBox IoU” in Table 3 computed by comparing with the ground truth bounding boxes? If so, is it possible for “BBox IoU” to penalize a predicted bounding box that is valid but different from any ground truth ones?
>
> To ensure a rigorous and standardized metric for bounding box accuracy, we chose to compute the IoU, rather than rely on participant ratings, which would be more subjective and participants may have different scales on the degree of accuracy for the bounding box. The ground truth bounding box for each design comment generated by the baseline and pipeline were manually determined by the authors, who agreed on each ground truth bounding box.
>
> Yes, it is possible  for “BBox IoU” to sometimes penalize a predicted bounding box that is valid but different from the determined ground truth bounding box, so the BBox IoU would actually be an underestimate to the bounding box accuracy.
>
>
> Please let us know if you have any further questions or would like to see additional changes.

---

> > ### Author Response · Authors · 2024-11-23
> > **Follow Up**
> >
> > Dear Reviewer PCXV,
> >
> > We are wondering if our rebuttal has addressed your concerns. Would you please get back to us on our responses?
> >
> > Thank you!

---

> > > ### Comment · Reviewer_PCXV · 2024-11-24
> > >
> > > Thanks for your rebuttal, which has addressed some of my concerns. However, I am still concerned about the limited amount of technical novelty involved in the paper, and thus choose to maintain my original rating.

---

> ### Author Response · Authors · 2024-11-27
>
> Dear Reviewer PCXV,
>
> Thanks for your further comments. We are glad that our rebuttal has addressed some of your concerns. It seems that the only concern you have now is regarding the amount of technical novelty, which we want to further address here.
> Our work is the first to bring iterative refinement to the multimodal domain. There are many considerations and experimentations with trial-and-error to find the right design for the set of prompting techniques in our approach. Critique generation is a complex multimodal task, and our technique clearly outperformed the baselines. Our findings are valuable for other researchers to continue investigating the topic, and our approach has also shown promise in solving multimodal tasks beyond critique generation. We believe simplicity is the strength of our approach as it makes the work easily reproducible by others, and applicable to other problems, while still incorporating many details and careful designs in our approach. We sincerely hope you can share our view.
>
> In the additional experiment and analysis that we conducted, which are already added to the revision, we found that our pipeline can also generalize to out-of-domain UIs, providing helpful design comments and bounding boxes to modern mobile UIs and websites using fewshot examples selected from only UICrit. Example outputs for these out-of-domain UIs are provided in Figures 11, 12, and 13 (Appendix). This finding further strengthens the practical utility of this pipeline.

---

### Official Review · Reviewer_JBj1 · 2024-11-03

**Soundness:** 2
**Presentation:** 2
**Contribution:** 2
**Rating:** 5
**Confidence:** 3

**Summary:**

This paper proposes a pipeline for generating design critique through interactions among large language models (LLMs), which provide design suggestions for UI interfaces and generate corresponding bounding boxes. By incorporating feedback techniques, the system refines bounding boxes or text based on the output of “Validation” step, resulting in more accurate text or bounding boxes. The authors also explore the model’s performance in tasks such as open vocabulary attribute detection and object detection.

**Strengths:**

1. The paper proposes Validation and Iterative Refinement modules that, depending on the specific context, selectively optimize either the box or the text, thereby further enhancing the model's accuracy.

2. The model outperforms previous works on the UICrit dataset.

**Weaknesses:**

1. **Minor Contributions**: Compared to previous work, the authors primarily added the Validation and Iterative Refinement modules; however, these modules lack novel and crucial design. Additionally, the IOU performance of the box generation module only reaches around 35%, limiting its practical applicability.

2. **Poor Result Presentation**: For both the design critique task and OVAD task, the paper provides minimal visual examples of the model's actual outputs. Instead, it predominantly uses ground truth images as illustrations, lacking clear demonstrations of the model's generated results.

**Questions:**

1. **Visualization of Results**

- **a)** Showcase as many successful and failed cases as possible. Provide a clear analysis of improvements compared to previous models and discuss reasons for unsuccessful cases.

- **b)** Visualize the refinement process with intuitive displays of box and text feedback, as well as refinement results. This will help validate the roles of the respective modules.

2. **Trainable VLM Choice** : Would directly fine-tuning advanced open-source VLM models, such as LLava or InternVL, on the UICrit dataset yield more accurate results?

3. **Simultaneous Generation of Box and Comment** : Since comments correspond to specific boxes, humans typically focus on the area (box) first and then generate corresponding text. Generating the text before locating the box seems counterintuitive. How would generating the comment and box simultaneously impact the effectiveness of the model?

4. **Ambiguous Metrics** : In both attribute detection and object detection tasks, mAP is a metric based on the percentage of successful samples, so performance differences are typically reported using absolute differences. However, in the paper, performance improvements are expressed as percentage increases in several sections, including the abstract, line 075, and line 483. This unconventional representation may introduce ambiguity.

---

> ### Author Response · Authors · 2024-11-19
>
> We thank Reviewer JBj1 for your insightful comments. We have revised the paper based on your feedback and uploaded the revised draft. We addressed your questions and concerns as follows:
>
> > Weakness 1: Minor Contributions: Compared to previous work, the authors primarily added the Validation and Iterative Refinement modules; however, these modules lack novel and crucial design. Additionally, the IOU performance of the box generation module only reaches around 35%, limiting its practical applicability.
>
> Re: “novel and crucial design”, we believe our paper introduces several novel designs of prompting methods, which are carefully examined in a complex multimodal task. For instance, we developed a new method for fine-grained visual grounding by iteratively zooming into an image enhanced with coordinate markers, and generating few-shot examples by incrementally perturbing the bounding box to simulate the refinement process. We added more examples illustrating how the refinement progresses in the Appendix of the revision (e.g., Fig 8 and 9). Overall. we designed the prompts for each module and evaluated them extensively. Our paper pushes the boundaries of what prompting alone can achieve for a complex multimodal task (UI critique generation). The paper provides insights into prompting methods for such tasks, such as whether prompting can improve upon existing baselines, how to implement it effectively, and whether the performance improvements from our method generalize to other multimodal tasks. We believe these make valuable research contributions for others to explore multimodal prompting in the future.
>
> Re: IOU performance, we want to clarify that, while the IoU of the box generation and refinement modules is around 35%, the IoU of the entire pipeline is actually 45.1%, according to the results from the human evaluation study (Table 3). This IoU improvement is due to the validation and text refinement modules.
>
> > Weakness 2: Poor Result Presentation: For both the design critique task and OVAD task, the paper provides minimal visual examples of the model's actual outputs. Instead, it predominantly uses ground truth images as illustrations, lacking clear demonstrations of the model's generated results.
>
> Thanks for pointing it out. We have addressed the presentation issue by including more generated results in the revision. See our response for Question 1 below.
>
> > Question 1: Visualization of Results
> >a) Showcase as many successful and failed cases as possible. Provide a clear analysis of improvements compared to previous models and discuss reasons for unsuccessful cases.
> >b) Visualize the refinement process with intuitive displays of box and text feedback, as well as refinement results. This will help validate the roles of the respective modules.
>
> We added Section A.4 in the Appendix that contains the results of a qualitative analysis. This section contains Figure 5, which presents two example outputs that demonstrate how our pipeline improves upon the baseline. These examples illustrate that our method reduces the occurrence of invalid, generic, and irrelevant design comments that are unhelpful, and produces more accurate, focused, and relevant feedback for the input screenshot and also produces more accurate bounding boxes. Figure 6 (Appendix) includes two examples where our pipeline performed worse than the baseline, where the pipeline eliminated a few valid design comments and generated a slightly less accurate bounding box. We have updated Section 5.4 with these additional qualitative results.
>
> Additionally, we have included Figure 8 in the Appendix, which visualizes the iterative refinement process for the bounding box based on a design comment, and Figure 9, which illustrates the refinement process for a design comment based on a bounding box. Both figures show the final results of their respective iterative refinement processes.

---

> ### Author Response · Authors · 2024-11-19
>
> (Continued From Above Comment)
> > Question 2: Trainable VLM Choice: Would directly fine-tuning advanced open-source VLM models, such as LLava or InternVL, on the UICrit dataset yield more accurate results?
>
> With sufficient data, fine-tuning would eventually outperform prompting. However, prompting requires much less data and computing resources. In this paper, we meant to push the boundary to investigate how far prompting can go for this complex multimodal task. That said, we have conducted additional fine-tuning experiments to make the paper more complete. We finetuned Llama 3.2 11b on UICrit for 3 epochs and have obtained the following results:
>
> | Metric | Finetuned Llama-3.2 11b | Gemini-1.5-pro | GPT-4o |
> | - | - | - | - |
> | Comment Similarity | 0.842 |  0.702  | 0.701 |
> | Estimated IoU | 0.230 | 0.199 | 0.275 |
>
> While finetuned Llama 3.2 has comparable Estimated IoU to our pipeline and higher comment similarity, qualitatively, it generates a very limited set of comments that cover text readability, visual clutter, and general comments on visual appeal. In contrast, our pipeline produces a more diverse range of critiques. We have updated Table 2 with these quantitative results. We have added Figure 7 in the Appendix, which provides example outputs from both the fine-tuned Llama model and our pipeline to illustrate this, and we added the results of this qualitative analysis to Section A.4 of the Appendix. We have also updated Section 5.4 and the Discussion (Section 7) with this new finding.
>
> > Question 3: Simultaneous Generation of Box and Comment : Since comments correspond to specific boxes, humans typically focus on the area (box) first and then generate corresponding text. Generating the text before locating the box seems counterintuitive. How would generating the comment and box simultaneously impact the effectiveness of the model?
>
> We agree that humans typically focus on the region (bounding box) before generating the corresponding critique. However, as we discuss in lines 203-207, we observed that LLMs often struggle with object grounding. If the model is asked to mark problematic regions first, it is more likely to produce inaccurate bounding boxes, leading to erroneous design comments based on those regions. This, in turn, increases the likelihood of errors in the subsequent refinement steps. We believe this finding will benefit others who will be designing prompts for multimodal tasks in the future.
>
> Regarding simultaneous generation of both text and box, Duan et al. (CHI 2024) found that dividing major tasks for LLMs into more specific subtasks improves performance. Therefore, we separated the design critique and bounding box estimation into separate subtasks, following the method by Duan et al. (UIST 2024).
>
> > Question 4: Ambiguous Metrics : In both attribute detection and object detection tasks, mAP is a metric based on the percentage of successful samples, so performance differences are typically reported using absolute differences. However, in the paper, performance improvements are expressed as percentage increases in several sections, including the abstract, line 075, and line 483. This unconventional representation may introduce ambiguity.
>
> Thank you for pointing out the confusing reporting of results. We have replaced the percent improvements with the absolute differences in the revision.
>
>
> Please let us know if you have any further questions or would like to see additional changes.

---

> > ### Author Response · Authors · 2024-11-23
> > **Follow up**
> >
> > Dear Reviewer JBj1,
> >
> > We are wondering if our rebuttal has addressed your concerns. Would you please get back to us on our responses?
> >
> > Thank you!

---

> ### Comment · Reviewer_JBj1 · 2024-11-24
>
> Thank you for your answers. However, in my opinion, given the limitations in the paper's innovativeness, its performance in terms of applicability is more highly anticipated. As such, more comprehensive visualization results are expected. For example:
>
> 1. **More usage examples are expected**: The current effective visualization examples in the paper are limited to about 4-5, which lack persuasiveness.
> 2. **Out-of-domain validation capability**: **The visualized UI interfaces in the paper seem somewhat outdated** (e.g., uniform electronic buttons at the bottom, application-related information shown in the figures, etc., resembling design styles from 7-8 years ago). Since the paper utilizes advanced models like GPT, more modern UI interfaces are anticipated.   In addition to, the ability to critique other systems' UIs and computer UI interfaces is also expected.
>
> If the corresponding capabilities can be demonstrated, I would be more than willing to adjust my score upward.

---

> > ### Author Response · Authors · 2024-11-26
> >
> > Thank you very much for your additional feedback, which has made our paper stronger.  We have revised the paper and uploaded a draft to address your feedback. We addressed your comments as follows:
> >
> > > 1. More usage examples are expected: The current effective visualization examples in the paper are limited to about 4-5, which lack persuasiveness.
> >
> > We have added 12 additional examples from the pipeline (Figures 5, 6, and 7 in the Appendix), which brings the total to 16-17 visualization examples. We have also updated Section A.4.1 (Appendix) with some qualitative observations.
> >
> > > 2. Out-of-domain validation capability: The visualized UI interfaces in the paper seem somewhat outdated (e.g., uniform electronic buttons at the bottom, application-related information shown in the figures, etc., resembling design styles from 7-8 years ago). Since the paper utilizes advanced models like GPT, more modern UI interfaces are anticipated. In addition to, the ability to critique other systems' UIs and computer UI interfaces is also expected.
> >
> > Yes, it is true that the UI screens in UICrit are older, as they were taken from the RICO dataset, which was collected in 2014. To evaluate the pipeline's performance on out-of-domain UIs, we used few-shot examples from UICrit and ran it on four modern Android screens (2024) sourced from Mobbin (https://mobbin.com/), four modern iOS screens (2024) taken from DesignVault (https://designvault.io/), and five modern websites (2024) taken from Mobbin, for a total of 13 UIs. The outputs are visualized in Figures 11, 12, and 13 (Appendix). We found that the pipeline was still able to generate helpful design feedback with reasonably accurate bounding boxes for these out-of-domain UIs. Notably, the results for the modern iOS UIs were comparable to those from the UICrit test split (Figures 5, 6, and 7 in the Appendix). We added Section A.4.2 in the Appendix that discusses these qualitative results.
> >
> > Please let us know if you have any further questions.

---

> > > ### Comment · Reviewer_JBj1 · 2024-11-27
> > >
> > > Thanks for authors' response. I will increase my score from 3 to 5. The reason I have not leaned towards acceptance is that the visualization results have not demonstrated an enough outstanding performance (e.g., the position of the boxes is still not accurate enough, especially OOD validation) and the novelty remains limited.

---

> > > > ### Author Response · Authors · 2024-11-27
> > > >
> > > > Dear Reviewer JBj1,
> > > >
> > > > Thank you very much for increasing your score! We agree with you that there is still room for improvement. On the other hand, the critique task itself is complex and challenging due to its multimodal and open-ended nature, and our approach has made tangible progress here. For example, previous works revealed that LLMs are unable to obtain high accuracy in providing design comments [3,4]. Our pipeline addresses this limitation by improving the accuracy over existing baselines, reducing the gap from experienced human designer’s performance by up to 50% according to human ratings, which is a substantial advancement on the topic.
> > > >
> > > > Regarding bounding box accuracy, our approach is aimed to address a challenging setup by relying only on pixels, but there are shortcuts that we can take to immediately enhance the accuracy if we relax this setup. For instance, the DOM tree representation of the UI will contain the exact bounding boxes of UI elements and element groups. This information could be used to refine the output bounding boxes for the elements/groups discussed in the critiques by finding the closest bounding box from the DOM tree via IoU comparison, distances between the bounding box centers and sizes, or utilizing an LLM for matching as was done in [1]. This DOM representation is available through the UI’s XML code, or the internal UI mockup representation available in design tools like Figma. In the case that the DOM tree is not available, we could use a screen object parser [2] to extract UI element and group locations from the screenshot.
> > > >
> > > > If you think it would be helpful, we could include a discussion of incorporating these enhancements into our pipeline for better accuracy in Section A.4 of the Appendix. Additionally, we could provide examples of refined bounding boxes using one of the techniques mentioned above.
> > > >
> > > > References:
> > > >
> > > > [1] Zhao, Haoyu et al. “LLM-Optic: Unveiling the Capabilities of Large Language Models for Universal Visual Grounding.” ArXiv abs/2405.17104 (2024): n. Pag.
> > > >
> > > > [2] Jason Wu, Xiaoyi Zhang, Jeff Nichols, and Jeffrey P Bigham. 2021. Screen Parsing: Towards Reverse Engineering of UI Models from Screenshots. In The 34th Annual ACM Symposium on User Interface Software and Technology (UIST '21). Association for Computing Machinery, New York, NY, USA, 470–483. https://doi.org/10.1145/3472749.3474763
> > > >
> > > > [3] Peitong Duan, Jeremy Warner, Yang Li, and Bjoern Hartmann. 2024. Generating Automatic Feedback on UI Mockups with Large Language Models. In Proceedings of the 2024 CHI Conference on Human Factors in Computing Systems (CHI '24). Association for Computing Machinery, New York, NY, USA, Article 6, 1–20. https://doi.org/10.1145/3613904.3642782
> > > >
> > > > [4] Yuwen Lu, Tiffany Knearem, Shona Dutta, Jamie Blass, Clara Kliman-Silver, and Frank Bentley. 2024. AI Is Not Enough: A Hybrid Technical Approach to AI Adoption in UI Linting With Heuristics. In Extended Abstracts of the CHI Conference on Human Factors in Computing Systems (CHI EA '24). Association for Computing Machinery, New York, NY, USA, Article 501, 1–7. https://doi.org/10.1145/3613905.3637135

---

> ### Author Response · Authors · 2024-11-28
> **Follow-up**
>
> As a follow-up to our previous comment, we tested the simple method that we discussed above, which utilizes the DOM tree to refine the generated bounding boxes through IoU comparison. This simple add-on to our approach noticeably improved the quality of the bounding box accuracy. We added Figures 14 and 15 (Appendix) in the latest revision to illustrate the new results and included Section A.4.3 in the Appendix to discuss this add-on and other potential refinement techniques that can enhance our method. This additional step could potentially be applied at the end of our pipeline to clean up the generated bounding boxes.

---

### Meta-Review · Area_Chair_wpjE · 2024-12-22

**Metareview:**

Although one of the reviewers suggested acceptance, others raised concerns regarding the limited novelty and insufficient analysis. Specifically, the reviewers agreed that the technical contributions were marginal. The AC also reviewed the paper, the reviews, and the rebuttal. The paper is well written (acknowledged by other reviewers), but iterative refinement has been widely explored in the literature, and applying this idea to the given task is considered incremental. Therefore, the AC agrees that the current version is not ready for acceptance. The AC strongly encourages the authors to address all the reviewers' concerns and resubmit to a future venue.

**Additional Comments On Reviewer Discussion:**

After rebuttal, the reviewers agreed that the paper has limited technical contribution.

---

### Decision · Program_Chairs · 2025-01-22

Reject